# Super-resolution imaging of synaptic and Extra-synaptic AMPA receptors with different-sized fluorescent probes

**Sang Hak Lee[1], Chaoyi Jin[1], En Cai[1†], Pinghua Ge[1], Yuji Ishitsuka[1], Kai Wen Teng[1], Andre A de Thomaz[1‡], Duncan Nall[1], Murat Baday[1], Okunola Jeyifous[2], Daniel Demonte[3], Christopher M Dundas[3§], Sheldon Park[3], Jary Y Delgado[2], William N Green[2], Paul R Selvin[1]\***

[1]Department of Physics, Center for Biophysics, and Quantitative Biology, and Center for the Physics of Living Cells, University of Illinois, Urbana-Champaign, Champaign, United States; [2]Department of Neurobiology, University of Chicago and the Marine Biological Laboratory, Chicago, United States; [3]Department of Chemical and Biological Engineering, University at Buffalo, Buffalo, United States

**Abstract** Previous studies tracking AMPA receptor (AMPAR) diffusion at synapses observed a large mobile extrasynaptic AMPAR pool. Using super-resolution microscopy, we examined how fluorophore size and photostability affected AMPAR trafficking outside of, and within, post-synaptic densities (PSDs) from rats. Organic fluorescent dyes ($\approx 4$ nm), quantum dots, either small ($\approx 10$ nm diameter; sQDs) or big (>20 nm; bQDs), were coupled to AMPARs via different-sized linkers. We find that >90% of AMPARs labeled with fluorescent dyes or sQDs were diffusing in confined nanodomains in PSDs, which were stable for 15 min or longer. Less than 10% of sQD-AMPARs were extrasynaptic and highly mobile. In contrast, 5–10% of bQD-AMPARs were in PSDs and 90–95% were extrasynaptic as previously observed. Contrary to the hypothesis that AMPAR entry is limited by the occupancy of open PSD 'slots', our findings suggest that AMPARs rapidly enter stable 'nanodomains' in PSDs with lifetime $\geq 15$ min, and do not accumulate in extrasynaptic membranes.
DOI: https://doi.org/10.7554/eLife.27744.001

**\*For correspondence:**
selvin@illinois.edu

**Present address:** [†]University of California, San Francisco, San Francisco, United States; [‡]Institute of Physics"Gleb Wataghin" University of Campinas, Campinas, Brazil; [§]Department of Chemical Engineering, University of Texas at Austin, Austin, United States

**Competing interests:** The authors declare that no competing interests exist.

## Introduction

A major goal of cellular and molecular neuroscience is to characterize how synapses change during synaptic plasticity, the leading cellular correlate of learning and memory (*Volk et al., 2015*; *Huganir and Nicoll, 2013*; *Kneussel and Hausrat, 2016*). The position and dynamics of the two major ionotropic glutamate receptors (iGluRs), α-amino-3-hydroxy-5-methyl-4-isoxazolepropionic acid receptors (AMPARs) and N-methyl-D-aspartate receptor (NMDARs), are critical for synaptic strength. AMPARs traffic intracellularly from sites of synthesis and assembly to exocytosis sites near synapses (*Anggono and Huganir, 2012*; *Henley et al., 2011*; *Greger and Esteban, 2007*). Despite much elegant work, it is still unclear what mechanisms limit AMPAR entry into and exit out of PSDs (*Malinow and Malenka, 2002*) (*Park et al., 2004*) (*Wang et al., 2008*). The different steps involved in AMPAR trafficking at synapses under constitutive conditions (no plasticity) are depicted schematically in *Figure 1a*. As shown, AMPAR trafficking consists of its exo- and endocytosis at sites on the plasma membrane (steps 1 and 1', respectively), diffusion to and from these sites and PSDs (steps 2 and 2') and entry into and exit from PSDs (steps 3 and 3'). The positioning of the AMPAR endo- and exocytosis sites with respect to PSDs is uncertain with some evidence for endo- and exocytosis sites within spines relatively close to PSDs (*Kennedy et al., 2010*; *Lu et al., 2007*), other evidence that

**eLife digest** Forgetting is a common experience in our everyday life. Yet much remains unknown about how we remember, and about why our memories sometimes fail us. The brain contains 80 to 100 billion nerve cells or neurons, which communicate with one another at junctions called synapses. At a synapse, one neuron releases a chemical message, which must diffuse across a small gap, and then activate proteins called receptors on another neuron. If the first neuron activates the second repeatedly, the second cell responds by inserting additional receptors into its membrane at the synapse. This strengthens the connection between the two neurons.

Strengthening of synapses is thought to be one of the key mechanisms underlying learning. To confirm this, it would be helpful to be able to monitor the movement and position of individual receptors at synapses. However, the space between the two nerve cells at at synapse, called the synaptic cleft, is no more than 40 nanometers wide. This is about 25 times thinner than a human hair, and too small to be seen with light microscopy. Electron microscopy can visualize synapses, but does not work in living tissue. The only other option is to attach a fluorescent label – either a dye or a man-made crystal called a quantum dot – to a protein found in synapses and monitor the resulting fluorescence. Though the probe must be small enough to pass through the synaptic cleft to do this.

Using fluorescence microscopy, researchers have examined the distribution in synapses of proteins called AMPA receptors, which have a key role in memory. Multiple studies have shown groups of AMPA receptors gathered outside synapses. This has led to the suggestion that during learning, AMPA receptors wait outside the synapse until a space becomes available within the synapse's membrane. However, this has yet to be confirmed directly, in part because conventional fluorescent dyes and quantum dots are too bulky to enter synaptic clefts when bound to a receptor.

Lee et al. have now developed a quantum dot that is only 10 nanometers wide and therefore small enough to enter the synaptic cleft with an AMPA receptor attached. These small quantum dots were then used to label AMPA receptors in neurons collected from rats and then grown in a petri dish, which provided a completely new view of synapses. The images show that the majority of AMPA receptors in neurons circulate within confined domains – a little like holding pens – inside the synapse, rather than waiting outside as previously assumed. Labeling the receptors with smaller 4-nanometer-wide fluorescent tags produces a similar picture. Further work is still need to determine how AMPA receptors get into the synapse and contribute to new memories.

DOI: https://doi.org/10.7554/eLife.27744.002

the sites are on dendrite shafts outside of spines (*Yudowski et al., 2007*; *Tao-Cheng et al., 2011*), and also that both locations may exist regulated by synaptic plasticity (*Tao-Cheng et al., 2011*). Another complication is that the recycling pathway under constitutive conditions may be different from the pathway with synaptic plasticity (*Zheng et al., 2015*; *Henley et al., 2011*). This suggests that different endo- and exocytosis sites may exist under constitutive conditions versus synaptic plasticity.

In the last decade, advanced fluorescence microscopic techniques have pioneered work on iGluR dynamics (*Heine et al., 2008*; *Groc et al., 2007*; *Nair et al., 2013*; *MacGillavry et al., 2013*; *Kneussel and Hausrat, 2016*). Commercial or 'big' quantum dots (bQDs) with a diameter >20 nm were observed (at least occasionally) within synaptic clefts of inhibitory synapses by electron microscopy (*Dahan et al., 2003*). bQDs are extremely bright, photostable and their position can be localized to within ~10 nm in the millisecond time-scale (*Pinaud et al., 2010*). Studies of AMPARs tagged with bQDs led to the conclusion that 50–80% of AMPARs are extrasynaptic and highly mobile (*Groc et al., 2007*; *Opazo et al., 2010*). This significant highly mobile pool of extrasynaptic AMPARs has been proposed to exist because of the slow rate that extrasynaptic AMPARs enter PSDs. This slow rate of motion and accumulation of extra-synaptic AMPARs is thought to occur because of the limited availability of 'slots' in PSDs (See *Figure 1a*, route 3) (*Shi et al., 2001*; *Opazo et al., 2012*). Slots are defined as placeholders for AMPARs in PSDs (*Malinow et al., 2000*). Once in PSD slots, AMPARs are generally assumed to be immobilized, although Nair et al. observed that AMPARs underwent constrained diffusion in what they defined as 'nanodomains', that is sub-synaptic regions where AMPARs are confined to (*Nair et al., 2013*). New slots may become available when AMPARs

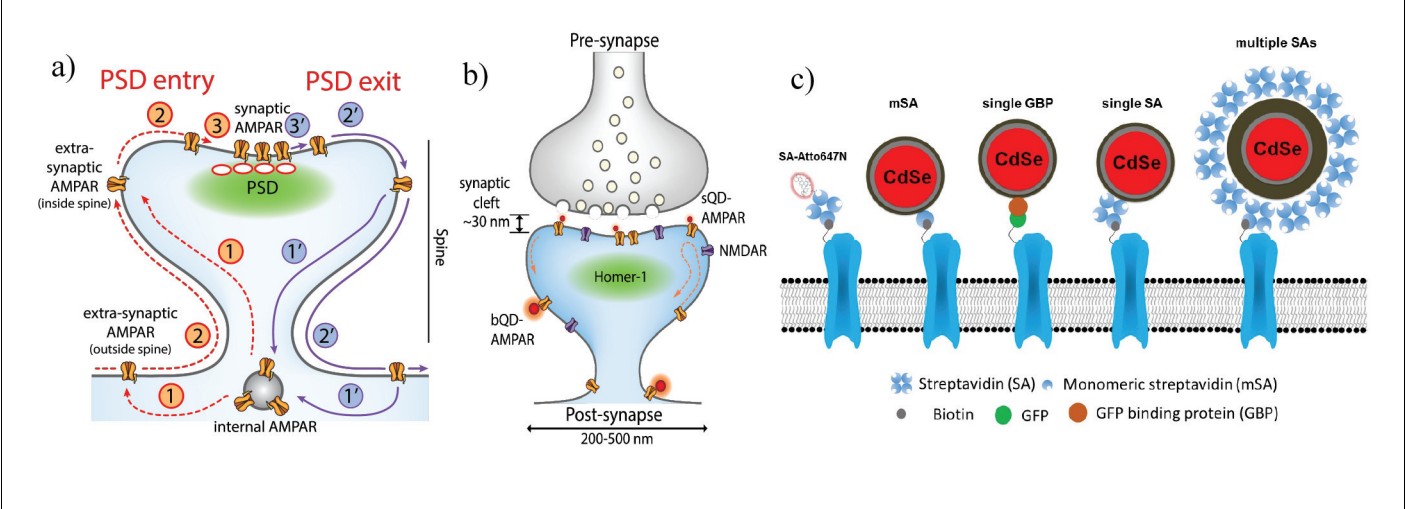

**Figure 1.** Schematic diagrams of the dynamics of AMPARs and labeling strategies for AMPARs. (a) AMPAR entry into and exit from the PSD. From an internal pool, AMPARs either enter the spine via membrane diffusion or through direct insertion (1) and then diffuse towards the PSD (2). From here, it enters the PSD in a rate-limiting step according to the slot theory (3). Similarly, the AMPAR may exit (1' – 3') the PSD in a similar manner. (b) Schematic of synapse labeled with a small (sQD) or big (bQD) quantum dots. Since the synaptic cleft width is only about 30 nm, the size of the fluorescent probe may sterically hinder entry into synaptic clefts. (c) Schematics of AMPA receptors with a variety of different sized probes and with various linkers. The smallest is the organic fluorophore linked by a Streptavidin (SA); next smallest in size are the small quantum dot linked by a monomeric SA (mSA), or an anti-GFP nanobody (called GBP) attached to a GFP-AMPAR, or a (single) SA; largest are commercial quantum dot which are attached to multiple SAs.
DOI: https://doi.org/10.7554/eLife.27744.003

exit from PSDs (route 3') or if more slots are created or removed by different forms of synaptic plasticity (*Malinow et al., 2000*; *Shi et al., 2001*; *Malinow, 2003*; *Giannone et al., 2010*) (*Groc et al., 2007*; *Heine et al., 2008*) (*Nair et al., 2013*) (*Makino and Malinow, 2009*) (*Opazo et al., 2012*) (*Henley and Wilkinson, 2016*). The presence of such a large pool of mobile AMPARs on the extrasynaptic surface of dendritic membranes is thought to be an important contributor to the mechanisms that underlie synaptic plasticity such as LTP and could help synapses recover from pair-pulse depression during periods of fast synaptic transmission (*Heine et al., 2008*; *Tao-Cheng et al., 2011*).

To avoid the potential problem of using large probes, other workers have used single particle tracking (spt) of small (<5 nm) fluorescent probes permanently (generally via a linkage protein) or temporarily bound to an AMPAR. One such technique is called sptPALM (spt- photoactivated localization microscopy) where an iGluR is fused with a photoactivatable fluorescent protein (pa-FP) (*Manley et al., 2008*). Another technique is called universal points accumulation for imaging in nanoscale topography (uPAINT) which measures an organic fluorescent probe that is temporarily bound to the receptor (*Sharonov and Hochstrasser, 2006*; *Giannone et al., 2010*). However, there are contradictory results using these methods. Using uPAINT, AMPARs were found to be highly mobile (*Giannone et al., 2010*), or stationary (*Nair et al., 2013*). Using sptPALM and FRAP, another fluorescence technique, workers found that AMPARs were highly mobile. However, fluorescent proteins for sptPALM and FRAP are sensitive to both internal and external AMPAR receptors, which may obfuscate the issue (*Nair et al., 2013*; *Heine et al., 2008*) (*Makino and Malinow, 2009*; *Ashby et al., 2006*).

Determining whether AMPARs or other neurotransmitter receptors are synaptic (outside PSDs) or extrasynaptic (inside PSDs) has been difficult because of four factors: (1) Synapses are nanometer in scale below the diffraction limit of visible microscopy; (2) Most microscopy has been limited to two-dimensions, where in reality, the dimensions are three-dimensional; (3) Recombinantly tagged imaging probes cannot readily distinguish intracellular vs. cell-surface receptors; (4) Probe photostability limits the localization precision and the length of image acquisition, particularly for single-molecule studies.

Here we use 3-dimensional super-resolution microscopy on cell-surface AMPARs labeled with different-sized fluorescent probes—bQDs (>20 nm), small quantum dots (sQD ≈10 nm) (*Cai et al.,*

*2014*) or organic fluorophore ($\approx$ 4 nm) in cultured neurons. These probes are highly fluorescent; bQDs are nearly infinitely photostable; sQDs are photostable for several minutes; and selected organic fluorophores can be photostable for ~1 min. We analyzed the diffusion constants, trajectories, distances from the center of PSDs, changes that occurred over time after transfection, and the role of the extracellular matrix in regulating diffusion. To better resolve PSD dimensions, we used 3-D super-resolution imaging of a PSD protein, Homer1c.

We find that > 90% of AMPARs labeled with organic fluorescent dyes or sQDs are found in PSDs. Less than 10% of these labeled AMPARs are highly mobile and extrasynaptic. However, when AMPARs are labeled with bQDs, virtually all AMPARs (90–95%) are highly mobile and extrasynaptic, presumably because bQDs are sterically hindered from entry into synaptic clefts. We conclude that mobile AMPARs enter PSDs and do not accumulate in extrasynaptic membranes as previously observed in studies using the AMPARs labeled with bQDs. Our data further suggest that AMPAR entry is not regulated by a limited number of 'slots' where AMPARs are bound after entry into PSDs. Instead AMPARs are found to diffuse within surprisingly stable 'nanodomains' confining AMPARs to sites within PSDs that may regulate access to presynaptic glutamate release.

## Results

### AMPAR labeling strategy and localization

We labeled surface-expressed tagged (AP-tag or EGFP) versions of the GluA2 subunits of the AMPAR receptor. They were transiently transfected into cultured rat hippocampal neurons and imaged at 13–16 days in vitro (DIV) (see Materials and methods, and [*Howarth et al., 2005*]). Three different fluorophores and ligands for coupling to AMPAR were used (*Figure 1c*): (1) a small organic fluorophore coupled to streptavidin (SA); (2) a sQD with three different ligands: a limiting amount of SA such that each sQD has approximately one SA; a 13kD single-chain antibody against GFP (from alpaca, commercially known as the Green Fluorescent Binding Protein, or GBP), which has only one target on a GFP$_{N-term}$-AMPAR (*Ries et al., 2012*; *Rothbauer et al., 2008*; *Wang et al., 2014*). (In one case, it has also been directed to a pHluorin-AMPAR, a GFP-derivative (*Ashby et al., 2004*). The result for this construct is the same as GFP$_{N-term}$-AMPAR); (3) a monomeric streptavidin (mSA), which has only one binding site for biotin and was further molecular engineered from a recent one (*Lim et al., 2013*; *Lim et al., 2011*; *Chamma et al., 2016*), in order to minimize non-specific labeling with neurons (Materials and methods and *Figure 4—figure supplement 1*). The SA and mSA were used with the same AMPAR constructs as used with the bQDs. The (tetrameric) SA was used at a ratio of one per sQD. A commercial bQD containing many streptavidins was also used (*Figure 1c*). Homer1c (tag with mGeos or mEos3.2) was co transfected to generate a super-resolution marker for PSDs. For all experiments, imaging was performed using a cylindrical lens for 3D reconstruction (*Huang et al., 2008*).

We noticed that most surface AMPA receptors labeled with SA-bQD one day following transfection (DIV14), reside in a widely spread *extra*-synaptic region (*Figure 2a and e*). This is consistent with our previous results (*Cai et al., 2014*) with the difference that the microscope stage drift and chromatic aberrations are corrected in this study. Chromatic aberration was corrected to with 4 nm (*Wang et al., 2014*). The drift was corrected to within approximately 2 nm in x and y, and 16 nm in z (*Lee et al., 2012*). (See also Materials and methods and *Figure 2—figure supplement 1–7*).

We also labeled AMPARs with a SA-sQD (which is approximately half the diameter of the bQDs or 1/8$^{th}$ the volume) and measured the distance of the AMPAR to the center of PSDs marked by Homer1c-mGeos. Qualitatively, the receptors are often in nanodomains, sub-synaptic regions where the receptor is confined (*Figure 2b, f*; also, Figure 4d and e). The distance between the center of Homer1c cluster to the center of the AMPAR-sQD is close: a maximum of 150±42 nm, with 75% and 91% of receptors being localized within 0.5 µm and 1.0 µm, respectively, of Homer1c.

Lastly, we also employed organic dyes to observe the diffusion of AMPA receptors in live neurons (*Figure 2c and g*; summarized in *Table 1*). This allows a direct comparison of AMPA receptors labeled with bQD and sQD and those labeled with organic fluorophores in live neurons. One advantage of the uPAINT technique using organic dyes compared to sptPALM is that the uPAINT technique only labels cell-surface receptors if cell-impermeable organic dyes are used. By choosing fairly photostable dyes, uPAINT also allows a relatively long time (limited by the photobleaching) for the

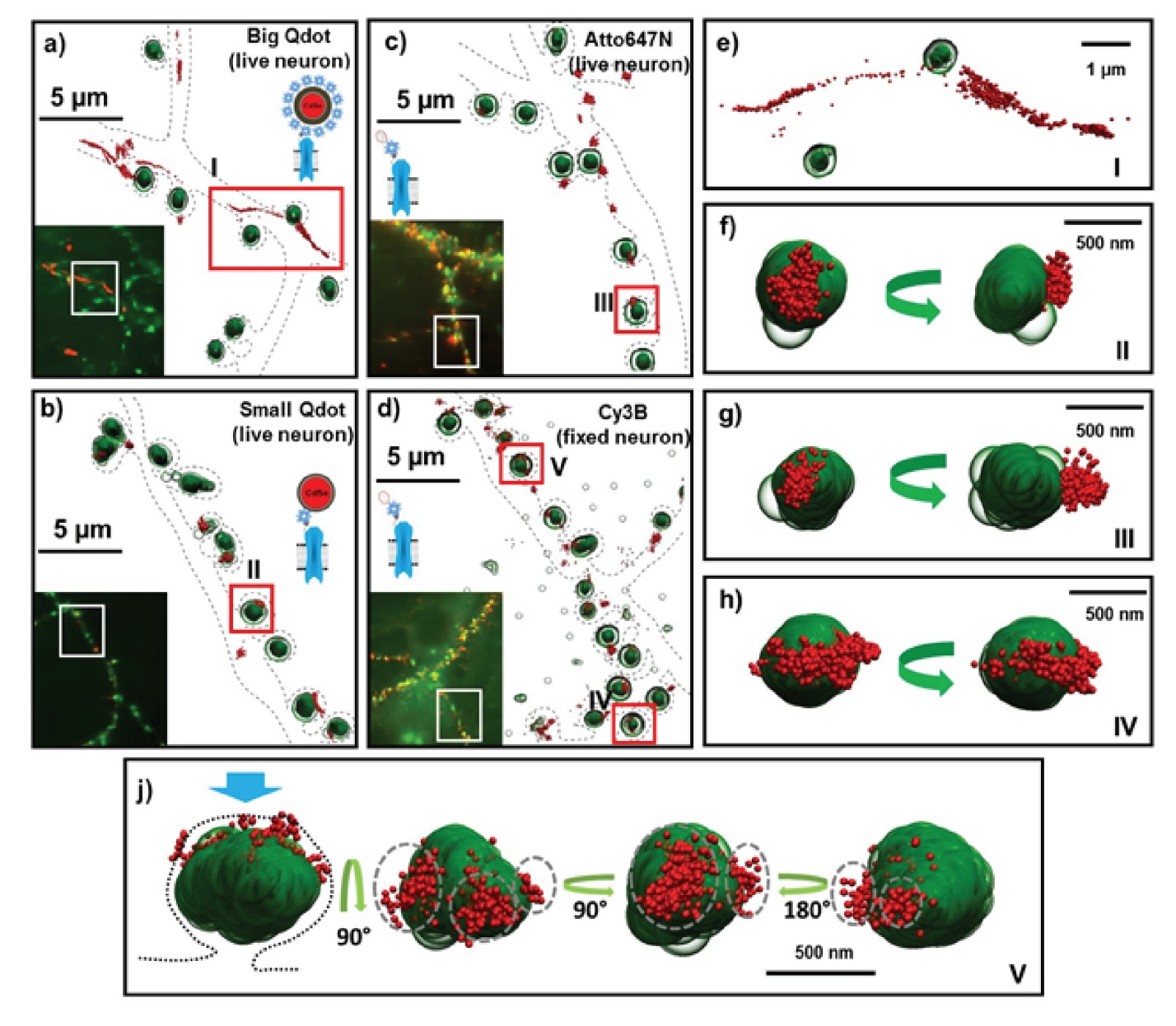

**Figure 2.** Fluorescent and super-resolution imaging of AMPARs and synapses. Big and Small quantum dots and organic fluorophores attached to AMPAR (red) on live cultured neurons with the PSD protein, Homer1c, labeled (green). Big (or commercial) quantum dots (**a, e**) bound to AMPA fail to get in the synapse and are spread extra-synaptically. Small quantum dots (**b, f**) and with small organic fluorophores (Atto647N) (**c, g**), bound to AMPAR, are stuck in nanodomains within the synapse. Fixing the neurons and using a Cy3 organic dye (**d**) can get the number of nanodomains per synapse; mostly there is just one (**h**), but occasionally there are multiple nanodomains (**j**). The dotted lines in a-d represent the morphology of dendrites. The green arrow (**f–j**) indicates the direction of synaptic cleft and the dotted line represents the shape of the synapse. Each image is the rotated structure of the synapse as represented by curved arrow. The gray dotted lines in j indicate the nanodomain of the AMPA receptors. For super-resolution images, the microscope stage drift and chromatic aberrations are corrected.

DOI: https://doi.org/10.7554/eLife.27744.004

The following figure supplements are available for figure 2:

**Figure supplement 1.** TEM image for Si mask which has fiduciary markers.

DOI: https://doi.org/10.7554/eLife.27744.005

**Figure supplement 2.** The upper panel represents cultured neurons on the poly-D-lysine (PDL) coated glass coverslip.

DOI: https://doi.org/10.7554/eLife.27744.006

*Figure 2 continued on next page*

*Figure 2 continued*

**Figure supplement 3.** Schematics for culturing neurons on the fiduciary marked coverslips and presentative images for correcting stage drift and chromatic aberration.
DOI: https://doi.org/10.7554/eLife.27744.007
**Figure supplement 4.** The difference of diffusion coefficient with and without correcting the stage drift.
DOI: https://doi.org/10.7554/eLife.27744.008
**Figure supplement 5.** Commercially available microscopes have a perfect focus system (PFS), which normally uses IR LED light, in order to reduce the drift along z-axis.
DOI: https://doi.org/10.7554/eLife.27744.009
**Figure supplement 6.** Traces of the stage drift and the precision of the correction.
DOI: https://doi.org/10.7554/eLife.27744.010
**Figure supplement 7.** Correcting chromatic aberration using nanohole pattern.
DOI: https://doi.org/10.7554/eLife.27744.011
**Figure supplement 8.** The precision of each fluorophores.
DOI: https://doi.org/10.7554/eLife.27744.012

motion to be measured. Here, we find that AMPARs labeled with organic-fluorophores on live cells are colocalized with Homer1c (*Figure 2c*). To measure the overall population of AMPARs within one synapse, as opposed to the behavior of an individual AMPAR, we obtained super-resolution (direct-STORM) images for neuronal receptors using a heavily labeled organic fluorophore (SA-Cy3B) after fixing the neurons (*Figure 2d and e*) (*Heilemann et al., 2008*). Cy3B is particularly good with STORM because it can be made photoactivatable with $NaBH_4$ (*Vaughan et al., 2012*). Furthermore, we can label extensively with many organic fluorophores because the hydrodynamic diameter of organic fluorophores is ~1 nanometer and even when conjugated with streptavidin, they are only about 4 nm in diameter (*Dikić et al., 2012*). Consequently, SA-organic dyes are significantly smaller than either bQDs or sQDs or even labeled-antibodies (which are commonly used to label AMPARs). In addition, non-specific labeling was insignificant. These findings were similar to those obtained using sparsely labeled sQDs and organic dyes labeling on live neurons. Thus, by all three techniques (with sQDs, with live- and fixed-neurons labeled with organic fluorophores) we find that the localization of AMPA receptors is predominantly at the synaptic regions. In addition, we observe in these fixed samples, that there is typically one nanodomain per synapse (*Figure 2h*), although multiple nanodomains do exist (*Figure 2j*).

## AMPAR-labeling: Diffusion constants and trajectory range

For all fluorescent tags, 'heat maps' were created to plot the diffusion constants (in $\mu m^2/s$) versus the AMPAR trajectory range (in $\mu m$) (*Figures 3–5*). In general, the lower left-hand corner of the map shows the diffusion of AMPARs where the trajectory range is limited, constrained by PSD (or slot) proteins (*Shi et al., 2001*; *Opazo et al., 2012*), and the diffusion coefficient of these receptors was slow; those in the upper right-hand corner are not constrained and move faster. Semi-quantitatively, we define 'slow' as $D \leq 10^{-1.75}$ (0.018) $\mu m^2/s$ and 'constrained' as a trajectory range of $\leq 10^{-0.1}$ (0.79)

**Table 1.** Summary of AMPA receptor diffusion that are synaptically-bound, measured with various fluorophores and tags.

| Fluorophore | Size (nm) | Tag | Diffusion coefficient ($\mu m^2/s$) | Trajectory range (nm) (±width) | Synaptic receptors (%) |
|---|---|---|---|---|---|
| Big Qdot | ~20 | SA | $10^{-2.5\pm0.02}$ | 390 ± 140 | 5 |
| Small Qdot | ~10 | SA | $10^{-2.41\pm0.02}$ | 418 ± 172 | 75 |
| | | GBP | $10^{-2.53\pm0.02}$ | 420 ± 170 | 82 (92) |
| | | mSA | $10^{-2.59 \pm 0.02}$ | 363 ± 144 | 88 (84) |
| Atto647 | ~5 | SA | $10^{-2.55 \pm 0.02}$ | 380 ± 156 | 77 (73) |
| CF633 | | | $10^{-2.70 \pm 0.01}$ | 380 ± 140 | 90 (93) |

Note: The first and second number for the synaptic receptors corresponds to the percentage of AMPAR in the lower left hand box and the area under the Gaussian curve for the trajectories, respectively, as outlined in each peaks of *Figures 3–5*. (The bQD data is for the 5% of all the AMPARs that are bound near a Homer1c, that is in the synapse. See discussion.)
DOI: https://doi.org/10.7554/eLife.27744.013

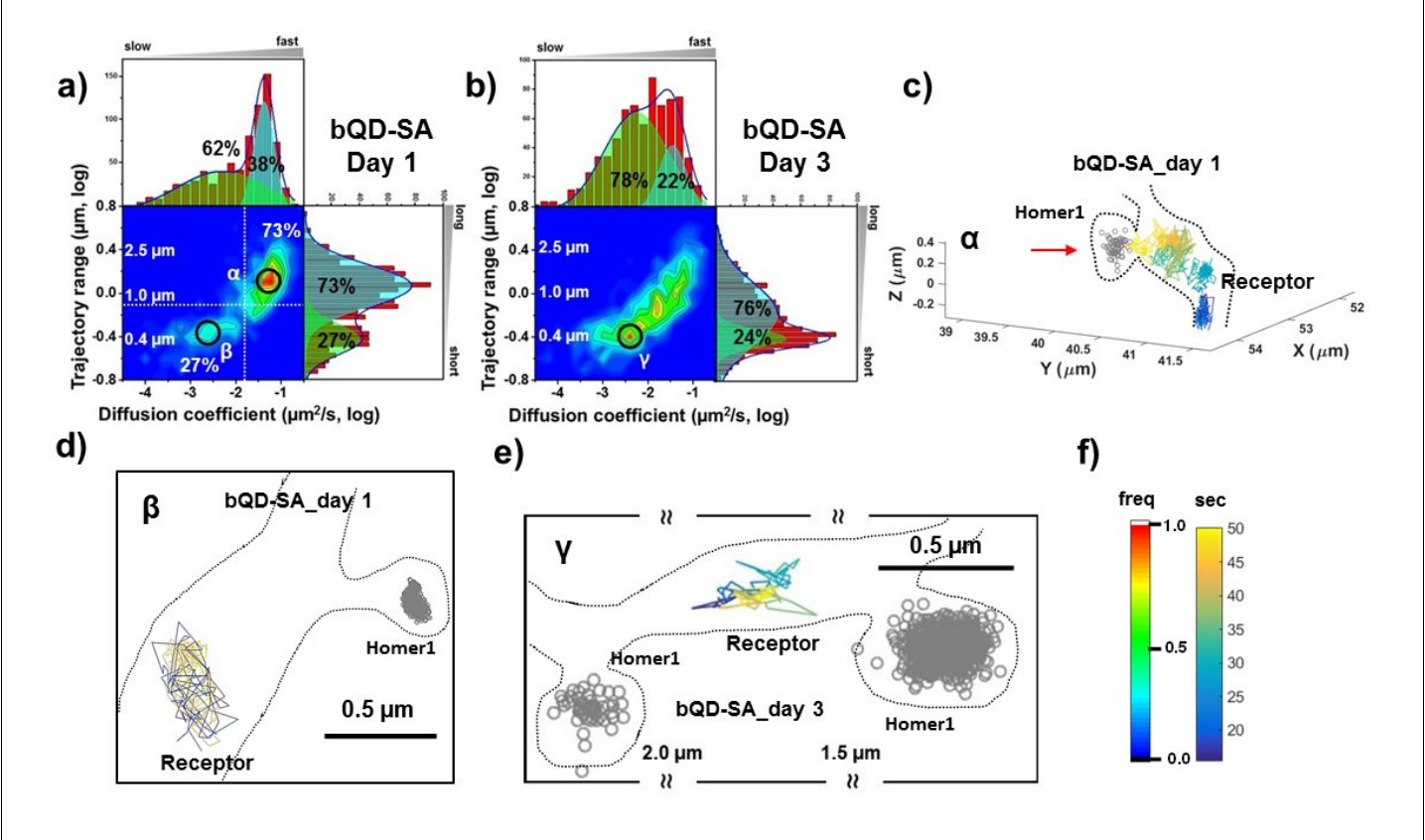

**Figure 3.** AMPARs are *not* synaptic as measured by bQDs. Two-dimensional plot of the diffusion coefficient and the trajectory range of AMPA receptors using big qdots (a) measured day 1 (n=483 traces, 8 cells, 4 independent dissociation, 1 dissociation per week) and (b) day 3 (n=715 traces, 4 cells, 4 independent dissociation, 1 dissociation per week) after transfection. White dot lines are for $10^{-1}$ µm²/s (0.79 µm) and $10^{-1.75}$ µm²/s (0.018 µm²/s)) for the trajectory range and the diffusion coefficient, respectively. Right upper side represents the population of fast diffusive receptors with longer trajectory and the left lower side represents the population of slow diffusive receptors with shorter trajectory. For all histograms, the distribution could be deconvoluted by Gaussian functions. Cyan colored Gaussian functions represent the fast diffusive receptors in diffusion coefficient and longer diffusive population in trajectory range. Green shaded Gaussian functions indicate slower and shorter diffusive receptors. On Day 1, the fast and long-distance peak (α) corresponds to ~73% of all molecules with ~$10^{-1.25}$ µm²/s (α) and ~1.3 µm, while the slow and short diffusion population (β) corresponds to ~27% of all molecules with ~$10^{-2.5}$ µm²/s and ~0.4 µm. On day 1, the α peak is not in the synapse but diffusive relatively freely along the synapse as represented in (c). Dotted lines in (c) represent the morphology of dendrite and the red arrow in c) indicates the synaptic cleft. The remaining 27% on day 1 in *Figure 3a* has a short trajectory and >10 x slower diffusion than the α peak. The majority of the slow diffusive receptors on both Day one and Day three are located away from synapse as in (d) and (e). At day three after transfection, the diffusion coefficient and trajectory range measured by using big qdots are significantly changed, probably due to crosslinking. (f) shows the color coding for population of heat maps and for time traces for a single receptor.

DOI: https://doi.org/10.7554/eLife.27744.014

The following source data is available for figure 3:

**Source data 1.** Diffusion coefficient and trajectory range measured by SA-bQD at day 1.

DOI: https://doi.org/10.7554/eLife.27744.015

**Source data 2.** Diffusion coefficient and trajectory range measured by SA-bQD at day 3.

DOI: https://doi.org/10.7554/eLife.27744.016

µm. These criteria were used as rough guidelines to be consistent with previous work (*Constals et al., 2015*) (*Cai et al., 2014*). The 2D heat maps can also be plotted as a function of DIV to see the effect of transfection time (see *Figures 3–5*). Alternatively, the 2D heat maps can also be turned into regular 1D histograms by compressing one axis and fitting the data to a series of Gaussian fits (see *Figures 3–5*). Importantly, because many cells were measured, (6–13 different culture plates from at least four independent culture preparations, corresponding to 483–1599 synapses), 2D histograms should be independent of absolute number of synapses (synaptogenesis).

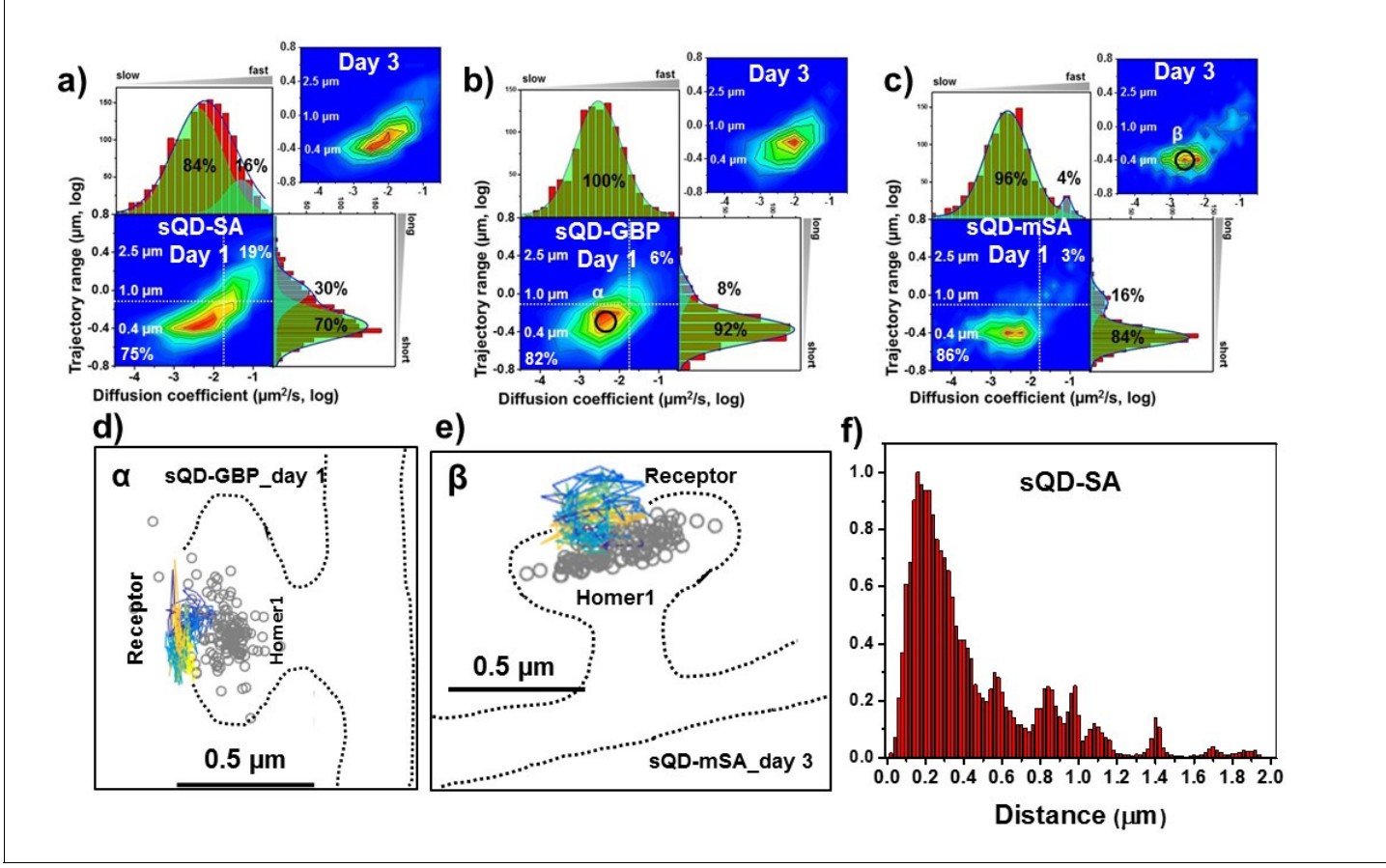

**Figure 4.** AMPARs are synaptic as measured by sQDs. Two-dimensional plot of the diffusion coefficient and the trajectory range of AMPA receptors using small qdots with various tags, (a) SA (n=1453 (11 cells, 6 independent dissociation, one dissociation per week) and 1523 (13 cells, 6 independent dissociation, 1 dissociation per week) traces for day 1 and 3, respectively), (b) GBP (attached to a GFP-AMPAR) (n=524 (5 cells, 4 independent dissociation, 1 dissociation per week) and 537 (7 cells, 4 independent dissociation, 1 dissociation per week) traces for day 1 and 3, respectively), and (c) mSA (n=750 (8 cells, 4 independent dissociation, 1 dissociation per week) and 701 (11 cells, 6 independent dissociation, 1 dissociation per week) traces for day 1 and day 3, respectively), measured day 1 and day 3 after transfection. White dot lines are for $10^{-1}$ µm²/s (0.79 µm) and $10^{-1.75}$ µm²/s (0.018 µm²/s)) for the trajectory range and the diffusion coefficient, respectively. Upper right side represents the population of fast diffusive receptors with longer trajectory and lower left size represents the population of slow diffusive receptors with shorter trajectory. Fitting the graphs via two Gaussian functions, the cyan represent the fast diffusive receptors (~$10^{-1.25}$ µm²/s) in diffusion coefficient and longer diffusive population (1 µm) in trajectory range; green indicates slower (~$10^{-2.5}$ µm²/s) and shorter diffusive receptors (~400 nm). For all three tags, 75–86% are slowly diffusive and have a short trajectory. Insets in a), (b), and (c) are two-dimensional plot for diffusion coefficient and trajectory range, measured day three after transfection. All insets are similar to the result, measured day 1. Slow diffusive receptors are mainly localized in synapses as in (d) (peak α in b) and (e) (peak β in c). Color-coding for heat maps and for time traces for a receptor is the same as *Figure 3f*. (f) histogram for the distance between receptors and the center of Homer1c. The histogram shows that 75% of receptors are localized within 0.5 µm of Homer1c (91% within 1.0 µm).
DOI: https://doi.org/10.7554/eLife.27744.017

The following source data and figure supplements are available for figure 4:

**Source data 1.** Diffusion coefficient and trajectory range measured by mSA-sQD at day 1.
DOI: https://doi.org/10.7554/eLife.27744.023

**Figure supplement 1.** Characterization of monomeric streptavidin and conjugation with small quantum dots.
DOI: https://doi.org/10.7554/eLife.27744.018

**Figure supplement 2.** (a) Agarose gel image for SA-, mSA-, and carboxylated- small qdots and (b) histogram for their size.
DOI: https://doi.org/10.7554/eLife.27744.019

**Figure supplement 3.** Measuring the diffusion coefficient of labeled qdots on the lipid bilayer in order to examine the possibility of cross-linking.
DOI: https://doi.org/10.7554/eLife.27744.020

**Figure supplement 4.** Measurement the run-length of kinesin using mSA-small qdot and SA-small qdot.
DOI: https://doi.org/10.7554/eLife.27744.021

**Figure supplement 5.** Free diffusive AMPA receptors labeled with mSA-sQD.

*Figure 4 continued on next page*

Figure 4 continued

DOI: https://doi.org/10.7554/eLife.27744.022

## AMPARs labeled with bQD coupled with SA are largely highly mobile and extrasynaptic

We used bQDs and were able to reproduce results from previous work showing large pools of receptors outside of the synapses. At post-transfection day one there are primarily two peaks in the heat map in *Figure 3a* with the majority (73% of all AMPAR, peakα) having a fairly fast diffusion constant ($10^{-1.25}$=0.056 $\mu m^2$/s) with about a ~ 1.3 $\mu m$ travel length, and a minor peak (27%, peak β) having slower diffusion constants ($10^{-2.5}$=0.0032 $\mu m^2$/s) and smaller trajectory ranges ($\approx$ 0.5 $\mu m$). As illustrated in *Figure 3c*, AMPARs in the α peak are localized outside the synaptic spine on the dendritic shaft distant from Homer1c labeling. Furthermore, the AMPARs are undergoing free diffusion confined over large (>1 um) regions in the dendritic shaft. The β peak has two components, both of which are diffusing slow. 5% of all AMPARs are colocalized with PSDs labeled by Homer1c; 22% are *not* localized to a labeled Homer1c (see *Figure 3d*). This 22% component could be confined extrasynaptic AMPARs, (possibly due to crosslinking), or they could be diffusing at PSDs not labeled by Homer1c.

On post-transfection day 3, the heat map in *Figure 3b* is significantly changed with a broader distribution of the diffusion coefficients and trajectory range, covering the range seen at post-transfection day 1. Unlike the result at post-transfection day 1, the population of free diffusion (i.e. the α peak) has significantly decreased, while the population of slow diffusion (i.e. the β peak) has increased. The data also show that the fraction of the slow diffusion is significantly increased on post-transfection day 3, as compared with that on post-transfection day 1. A representative trajectory is shown in *Figure 3e*, which shows the peak γ of the bQD undergoing limited-diffusion in a small region far from synapses similar to the β peak in *Figure 3a*.

For the bQDs, we measured AMPAR diffusion as a function of DIV (*Figure 3a* vs. b). All labels were examined after 1 day and after 3 days, following transfection, corresponding to DIV14 and DIV16, respectively. Because we observed a change in mobility we reasoned that this could be due to cross-linking of AMPAR by the multivalent bQD, or due to an increase in the amount of extracellular matrix. Both factors are known to influence the diffusion of surface-expressed AMPARs (*Chamma et al., 2016*; *Frischknecht et al., 2009*). (This result is further described in *Figure 6*.) Regardless of the exact reason, the fact that labeling with bQDs is a function of the number of DIVs is problematic.

## AMPARs labeled with sQD by different linkers (SA, GBP, or mSA)

When using sQDs, we found no accumulation of large pools of extra-synaptic receptors. Instead, we found that motion is largely constrained-diffusion in PSD nanodomains. The heat maps for the AMPARs labeled with sQDs (*Figure 4*) are strikingly different from the heat maps for AMPARs labeled with bQDs (*Figure 3*). First, the diffusion with any of the three different linkers resulted in much slower diffusion, with a diffusion coefficients ~ $10^{-2.5}$ $\mu m^2$/s. Results were different depending on which linker was used with the mean diffusion constant varying from $10^{-2.41\pm0.02}\mu m^2$/s for the SA linker (*Figure 4a*), which is largest and multivalent (*Figure 1b*), $10^{-2.53\pm0.02}\mu m^2$/s for the GBP linker (*Figure 4b*), which intermediate in size and monovalent and $10^{-2.59 \pm 0.02}\mu m^2$/s for the mSA linker (*Figure 4c*), the smallest and monovalent. These results suggest that both the size as well as the valency of the probe can influence the measurements.

More specifically, these trends yield 75% (70%), 82% (92%) and 84% (86%) slowly moving, that is *a vast majority of AMPAR-sQDs display constrained diffusion*. (The numbers in parenthesis correspond to taking the integrated area under the Gaussian fit). By looking at the diffusion constants, one sees a distribution of constrained diffusion: 84, 100% and 96%. Both results are in direct contrast with the results for labeling with the bQDs. In addition, the diffusion constant for AMPARs linked via SA to sQD is somewhat broader than the sQD coupled to GFP via GBP, which is somewhat broader than the mSA. (The mSA result is approximately equal to the smaller organic fluorophores—see below). This possibly implies that, although SA-sQDs are significantly better than the bQD to label

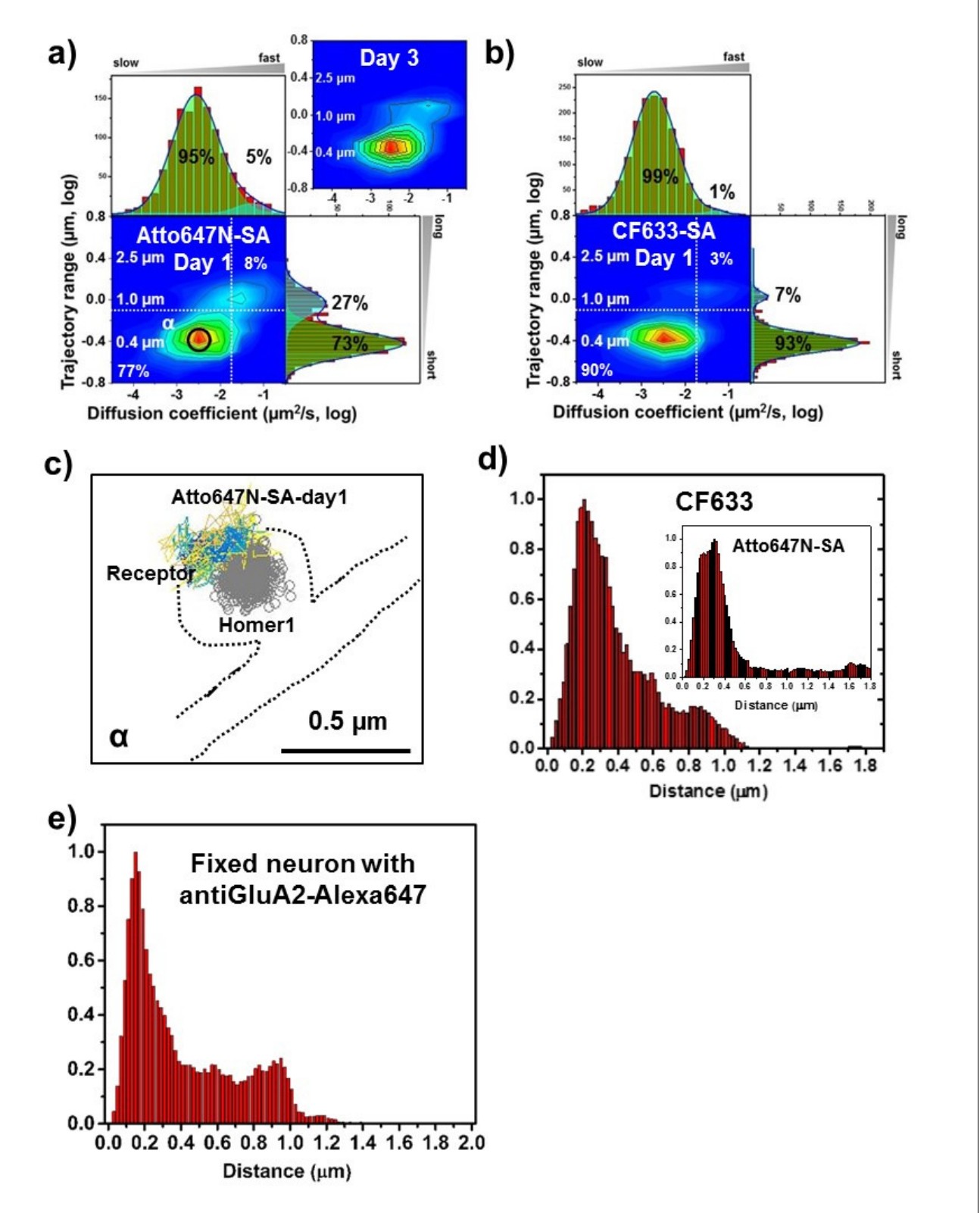

**Figure 5.** AMPARs are synaptic as measured by organic fluorophores. Two-dimensional plot of the diffusion coefficient and the trajectory range of AMPA receptors using 1 nM organic dyes for labeling, (a) Atto647N measured at day one as well as day 3 (n=1152 (9 cells, four independent dissociation, one dissociation per week) and 3063 (7 cells, three independent dissociation, one dissociation per week) traces for day 1 and 3, respectively) and (b) CF633 measured at day 1 (n=1599 traces, 11 cells, four independent dissociation, one dissociation per week). White dot lines are

*Figure 5 continued on next page*

*Figure 5 continued*

for $10^{-1}$ μm²/s (0.79 μm) and $10^{-1.75}$ μm²/s (0.018 μm²/s)) for the trajectory range and the diffusion coefficient, respectively. Upper right side represents the population of fast diffusive receptors with longer trajectory and lower left size represents the population of slow diffusive receptors with shorter trajectory. Cyan colored Gaussian functions represent the fast diffusive receptors (~$10^{-1.25}$ μm²/s) in diffusion coefficient and longer diffusive population (1 μm) in trajectory range. Green shaded Gaussian functions indicate slower (~$10^{-2.5}$ μm²/s) and shorter diffusive receptors (~400 nm). For both organic dyes, more than 90% have a small diffusion constant and have a 73–93% have a short trajectory (based on the diffusion coefficient graph). (c) We find that they are moving slowly next to the Homer1c. (d) histogram for the distance between receptors and the center of Homer1c on a live neuron labeled with AMPAR-SA-CF633 or Atto647N inset. The histograms shows that 81% and 78% of receptors, respectively, are localized within 0.5 μm. The color-coding for heat maps and for time traces for a receptor is the same as *Figure 3f*. (e) Histogram of distance between native AMPAR (labeled with antibody-Alexa647 against GluA2) and Homer1c-mGeos, and fixed. Here 71% of the receptors are within 0.5 μm of the Homer1c (96% within 1.0 μm), very similar to (d).

DOI: https://doi.org/10.7554/eLife.27744.024

The following source data and figure supplements are available for figure 5:

**Source data 1.** Diffusion coefficient and trajectory range measured by SA-CF633 at day 1.

DOI: https://doi.org/10.7554/eLife.27744.029

**Figure supplement 1.** Measurement of the photobleaching time of SA-atto647N.

DOI: https://doi.org/10.7554/eLife.27744.025

**Figure supplement 2.** Two dimensional heat maps for the diffusion coefficient and trajectory range measured at various concentrations (0.1, 0.3, 0.5, and 1.0 nM) of SA-Atto647N.

DOI: https://doi.org/10.7554/eLife.27744.026

**Figure supplement 3.** Measured the distance between the receptors and the center of Homer1c.

DOI: https://doi.org/10.7554/eLife.27744.027

**Figure supplement 4.** Image AMPA receptors using the antibody of GluA2.

DOI: https://doi.org/10.7554/eLife.27744.028

AMPARs, smaller ligands (like mSA) or organic-fluorophores (coupled via SA: see below) are better to study synaptic receptors in neurons.

We find that labeling with sQD to AMPAR is not a function of DIV, as day 14 is essentially identical to day 16, for all three ligands used. This says that the diffusion time and trajectory is a *not* a function of time, and therefore that the time-dependence, possibly due to cross-linking and differential interaction with extracellular matrix, is not a problem for sQDs-AMPARs. This gives us confidence about our results with sQDs.

We note that heat maps tell about the range of diffusion of AMPAR, but do not give the absolute distance from the PSD. Rather it is the trajectory range of receptors which does this (*Figure 2e–j*, or *Figure 3c–e*, or *Figure 4d–g*). To get whether the AMPAR is labeling near or in the PSD, we plot in *Figure 4f*, the histogram of the distance of AMPAR from the center of Homer1c. The histogram shows that 75% of receptors are localized within 0.5 μm of Homer1c and 91% within 1.0 μm, generally undergoing constrained-diffusion (*Figure 2d, f*). This strongly implies that the great majority of AMPARs are within the PSD. This further implies that our labeling strategy using the small probes does not limit access to the synaptic cleft.

We also need to take into account those sQDs that are bound to AMPARs but not in a synapse. (For the bQDs, this was 22–5%=17%.) We find that there is 12% of labeled mSA-sQD-AMPAR undergoing constrained diffusion not around a labeled Homer1c. Whether this is due to AMPAR constrained within a PSD containing an unlabeled Homer1c, or constrained within some other domain, is not known.

## AMPARs labeled with organic dyes coupled with SA

These results are similar to AMPARs labeled with sQDs. We labeled AMPARs using SA-Atto647N or SA-CF633 (*Figure 5*). Both of these dyes are bright, especially when several dyes are conjugated per streptavidin, and fairly photostable,>50 s observation time (*Figure 5—figure supplement 1*). (Why we get this enhanced photostability is unclear.) In addition, Atto647N has a positive charge and is relatively hydrophobic, whereas CF633 is negatively charged and hydrophilic (*Zanetti-Domingues et al., 2013*). Somewhat better, more specific labeling was shown with CF633. Overall, these factors have been found to be significant in terms of the measurement of the diffusion of membrane proteins (*Zanetti-Domingues et al., 2013*).

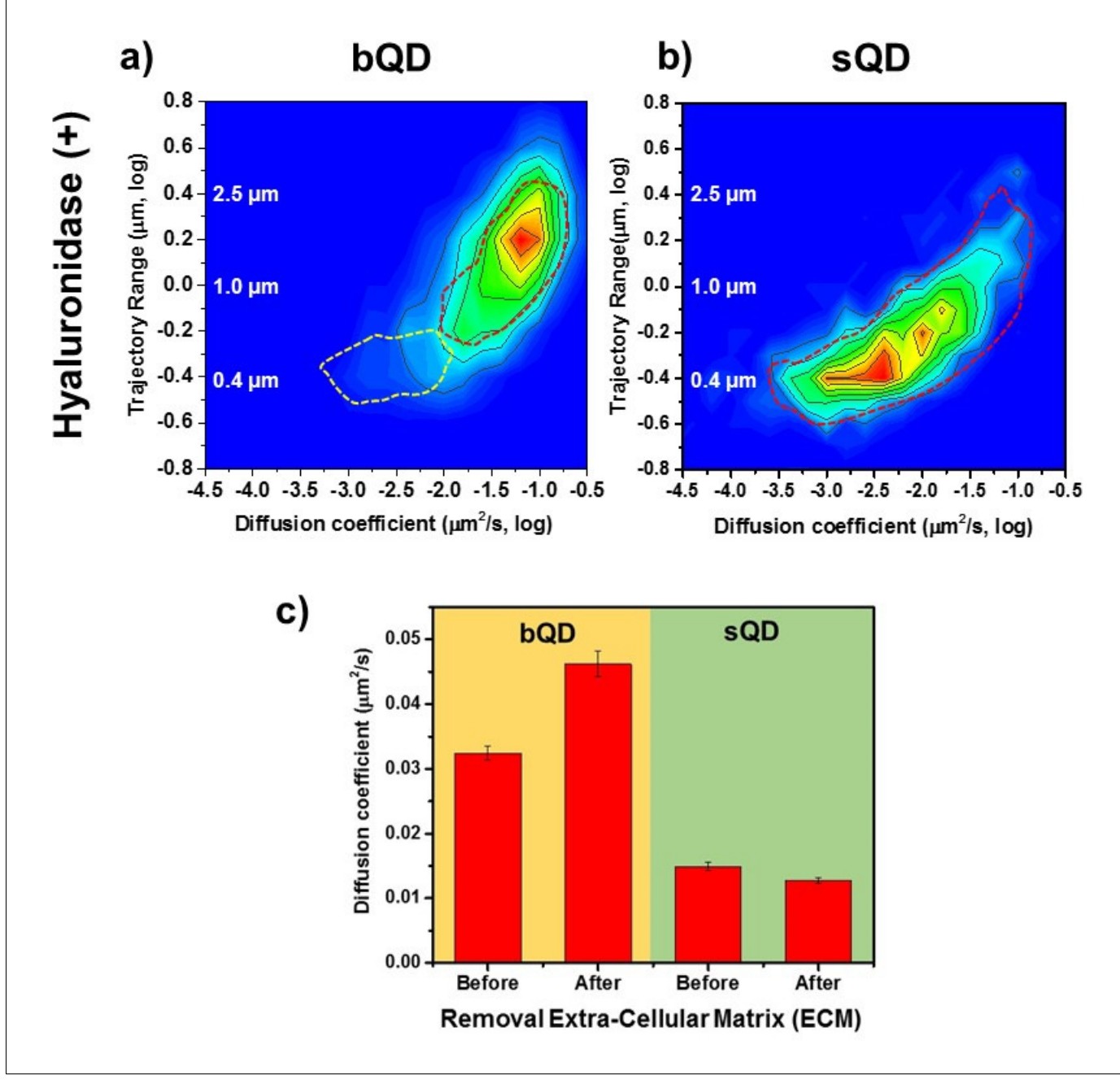

**Figure 6.** Lateral Diffusion of Synaptic AMPAR is Unaffected by Removal of ECM. Measurement the diffusion coefficient and trajectory range of the AMPA receptors using bQD-SA and sQD-SA after enzymatically removing extra-cellular matrix (ECM) using Hyaluronidase (100 unit/ml). (a) and (b) are 2D plots for diffusion coefficient and trajectory range measured with big qdots (n=2566 traces, 6 cells, 4 independent dissociation, 1 dissociation per week) and small qdots (n=1103 traces, 6 cells, 4 independent dissociation, 1 dissociation per week), respectively, after removal ECM. Dot lines circles represent the diffusion coefficient and trajectory range of Hyaluronidase (-) (taken from *Figure 3a* and *Figure 4a*, respectively). When using big qdots, the population of slow diffusion and short trajectory range disappears with treating with Hyaluronidase. However, with sQDs, there is no significant difference. The color-coding for heat maps is the same as *Figure 3f*. (c) The average diffusion coefficient, measured by using big qdots, after the removal of the ECM increases 1.4 fold from that at the condition of Hyaluronidase(-). On the contrary, the average diffusion coefficient at Hyaluronidase (+) is almost the same as without treating Hyaluronidase.

DOI: https://doi.org/10.7554/eLife.27744.030

A heat map shows that the diffusion constants and range of AMPARs are both quite small when labeled with Atto647N (post-transfection day 1 and day 3) and CF633 (post-transfection day 1) (*Figure 5a and b*). The results for Atto647N at various concentrations (0.1–1 nM) are similar (*Figure 5—figure supplement 2*). There is a slight difference between the two dyes chosen, which indicates that CF633 might have more specific labeling. With Atto647N, fitting the trajectories regardless of the diffusion constants, there is 27% of the AMPAR which has longer trajectory range and larger diffusion constants (*Figure 5a*); these are not much present with the CF633 dye, where we find 93% are constrained in nanodomains (*Figure 5b*). Finally, *Figure 5c* is the image of the main peak, α in *Figure 5a*, which shows that AMPAR is not just constrained, but is constrained around the PSD. *Figure 5d* indicates that 81% of AMPAR-SA-CF633 is localized within 0.5 μm of Homer1c, and this number increases to 98% within 1 μm. Similarly, for Atto647N, 78% is within 0.5 μm and 84% within 1 μm of Homer1c.

For fixed neurons, labeled with Cy3B, the results are similar *Figure 5—figure supplement 3*): 85% of the receptors are localized within 0.5 μm of Homer1c. For any of the dyes chosen, whether fixed or live, they indicate a close association between the Homer1c and AMPARs.

## Endogenous AMPARs labeled with organic dyes coupled to anti-GluA2 in non-transformed fixed neurons

We imaged native AMPARs labeled with antibody(GluA2)-Alexa647 and measured the distance between native AMPARs and Homer1c clusters using fixed neurons which we transfected with Homer1-mGeos. As shown in *Figure 5e* and *Figure 5—figure supplement 4*, the results show that 71% of AMPARs are localized within 0.5 μm, and 96% within 1 μm. This is very similar to the Atto647N and CF633 shown in *Figure 5d* (78-98%). The results using endogenous AMPARs labeled with antibodies indicate that the overexpression of GluA2 does not significantly affect the measurements.

## Removing extra-cellular matrix

The extra-cellular matrix (ECM) is a dense meshwork structure in surrounding brain cells (*Syková and Nicholson, 2008*). In neurons, surface receptors such as AMPARs and NMDARs, possibly interact directly or indirectly with the ECM, so that the ECM may affect the lateral diffusion of surface receptors in the synapses and surrounding locations (*Dityatev and Schachner, 2003*). Recent studies have shown that the ECM is also involved in the mechanism of recruiting glutamate receptors through lateral diffusion (*Dityatev and Schachner, 2003*; *Dityatev et al., 2010*; *Frischknecht et al., 2009*) (*Frischknecht and Gundelfinger, 2012*). In particular, one recent study claimed that when the ECM was enzymatically removed, the extra-synaptic AMPA receptors moved at a higher diffusion rate and the exchange of synaptic receptors was increased (*Frischknecht et al., 2009*). They therefore concluded that the ECM might regulate short-term synaptic plasticity.

However, the observations were on AMPARs labeled by bQDs, so their conclusions should be re-examined in light of our results above. We therefore compared AMPARs labeled with bQDs and sQDs on untreated cultured neurons and on neurons in which ECM was removed using Hyaluronidase. Recall that with bQD, in the presence of ECM (*Figure 3a* and dotted lines in *Figure 6a*), we have two populations of AMPAR-bQD: 73% are fast moving (D ~ $10^{-1.20 \pm 0.01}$ = 0.063 μm$^2$/s) over a large distance and 27% are slow moving (D ~ $10^{-2.0}$ - $10^{-3.0}$ μm$^2$/s), and constrained to be approximately 0.5 μm from the Homer1C. After ECM removal, we find that the slow-moving ones (the β peak in *Figure 3a* or the white dotted line in *Figure 6b*) are gone. All of the remaining AMPAR-bQDs freely diffuse outside of synapses on the membrane of dendrites. In contrast, with small qdots, we observe that the average and distribution of the diffusion coefficient of AMPA receptors was almost the same before ($D_{ave}$=0.015 ± 0.0006 μm$^2$/s) (*Figure 6b*, dotted line) and after ($D_{ave}$=0.013±0.0004 μm$^2$/s) ECM removal (*Figure 6c*). Thus, sQD-tagged AMPAR diffusion is not significantly affected by the presence of ECM, presumably because virtually all of these AMPARs are confined to PSDs where diffusion is limited by interactions with PSD scaffold proteins. Only the bQD-tagged AMPARs, which are virtually all extrasynaptic, are significantly affected by ECM. These results provide an explanation for the changes that occur in the diffusion and trajectories of the bQD-tagged AMPARs going from DIV 14 to 16 in *Figure 3a and b*. As the neurons develop in

culture, progressively more ECM is deposited on the neuronal membrane going from DIV 14 to 16 thereby reducing AMPAR diffusion and trajectories.

## Using sQDs for longer measurements of AMPARs in PSDs

We find that AMPARs labeled with sQDs differ from AMPARs labeled with bQDs in being able to consistently enter synaptic clefts and nanodomains within PSDs. We observe the same behavior with AMPARs labeled with organic fluorophores. The feature that distinguishes sQDs from organic fluorophores is the significantly longer photostability of the sQDs compared to the ≈ 1 min lifetime with the organic fluorophores. Using AMPARs labeled with sQDs we examined the stability of AMPAR nanodomains in PSDs over 15 min, exciting the sQDs for 30 s at a time, every 5 min. A previous study observed that most nanodomains were stable for up to one hour, but some nanodomains were dynamic, appearing and disappearing within minutes (*Nair et al., 2013*). This other study, however, used AMPARs labeled with either organic fluorophores or recombinant fluorescent proteins, which limited the imaging to the ensemble average of AMPAR in PSDs. They were therefore unable to observe the dynamics of single receptors except for much shorter times (<<minute), corresponding to the photobleaching time of organic fluorophores. Here we used sQDs to study the dynamics of the nanodomains of the same AMPAR in synapses for long periods (from 15 min repeatedly, and up to 60 min occasionally) of time. A total of 126 synapses were observed.

As shown in *Figure 7*, we first located AMPARs for 30 s, immediately followed by PSD imaging using Homer 1 c super-resolution. AMPARs were then imaged three times more, every 5 min for 15 min, and PSD imaged one final time (*Figure 7a*). The finite photostability of the fluorescent protein prevented the Homer1c from being monitored more than this. The dendrite morphology was also imaged using a transfected near-IR FP, which fills the cytoplasm of the dendrite (*Figure 7b*).

The nanodomain of the AMPAR, which was ~ 200 nm, remained in approximately the same position in the synapse for the entire fifteen minutes. As shown in *Figure 7c and d*, when imaged at the beginning (AMPAR-red, Homer1c-green) and the end of the fifteen-minute period (AMPAR-blue, Homer1c-brown), the synapse itself had changed position only slightly. To see how much the synapse had drifted over the 15 min, we aligned the images. We first made a linear adjustment, moving both the AMPA receptor trace and the Homer1c cluster about 100 nm and we then rotated the trace and cluster 19° to align them (*Figure 7c*) from one perspective and from then rotated the synapse 10°, perpendicular to the first axis, as shown in *Figure 7d*. With these relatively minor adjustments, the two images of the synapse nearly perfectly overlapped (right hand side of *Figure 7c, d*). In addition, *Figure 7e* shows that the shape of the trace of the AMPAR (~200 nm in size) observed at 0, 5, 10 and 15 min, remained quite constant. This indicates that the nanodomain, defined by the constrained area of a single AMPAR, was extremely stable for the fifteen-minute measurement period.

Occasionally, AMPAR nanodomains were observed to move within PSDs (*Figure 7—figure supplement 1*). Using the same protocol, the AMPAR nanodomain was first observed at the very edge of the PSD (0 min) after which the nanodomain moves further within the PSD and does not move further. As shown in *Figure 7—figure supplement 1(b–d*, the shape and size of the AMPAR nanodomain changes little (*Figure 7f*), and only the nanodomain position changed after the initial timepoint. In one example we also imaged a highly mobile extrasynaptic AMPAR enter into a nanodomain in PSD (*Figure 7—figure supplement 2*). As shown, the AMPAR during the initial time point is widely diffusing near a spine but outside of the PSD. During the last time 15 min time-period, the same AMPAR enters a nanodomain at the edge of the PSD. These images suggest that AMPARs immediately enter nanodomains as they arrive at PSDs after which the nanodomain can reposition within the PSD to a more stable location. In summary, due to the photostability of sQDs we have been able to characterize new details of AMPAR nanodomains and find that they are stable for significant times.

## Discussion

With synaptic plasticity, the numbers of AMPARs in PSDs change. Even under basal conditions, the AMPARs are recycling and therefore constantly moving in and out of synaptic clefts and PSDs. AMPAR endocytosis and exocytosis is very likely ongoing at synapses (*Greger and Esteban, 2007*) and occurs at sites outside of spines (*Anggono and Huganir, 2012*) or within spines away from PSDs (*Henley et al., 2011*). Consequently, it is important to identify where highly mobile AMPA

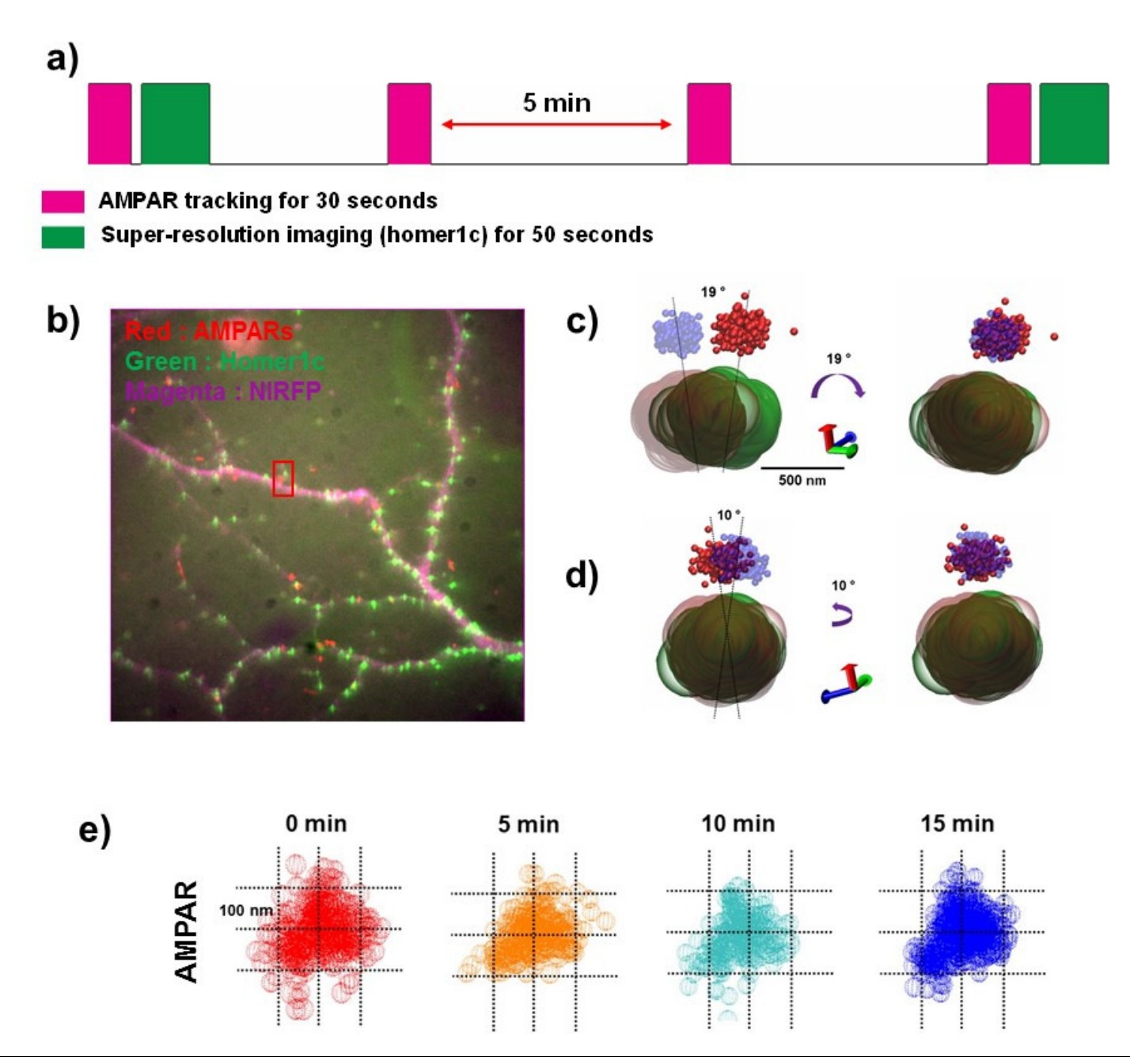

**Figure 7.** The dynamics of the AMPA receptor domain in synapses. Example of AMPAR and Homer1c moving very little over 15 min. (a) Schematics of long-term measurement dynamics for the AMPA receptor domain. AMPA traces were recorded for 30 s (50 ms exposure time) every 5 min (0 min ~15 min) and super-resolution images for synapses were taken at the beginning and the end of the 15 min period. (b) fluorescence image for synapses (homer1c-mGeos, green), AMPARs (sQD, red), and cell morphology (free-NIRFP, magenta). (c) and (d) show the difference of the position of an AMPAR and a synapse (Homer1c) from 0 min (AMPAR-red, Homer1c-green) to 15 min (AMPAR-blue, Homer1c-brown) at two different direction. At 0 through 15 min, when AMPAR and Homer1c are rotated 19° in a clockwise direction as shown in c) and 10° in a counterclockwise direction as shown in d), they overlap. (e) shows a stable nano-domain throughout the 15 min period. Red (0 min), orange (5 min), cyan (10 min), and blue (15 min) balls represent the AMPA receptor at various points in the trace of that AMPA receptor.

DOI: https://doi.org/10.7554/eLife.27744.031

The following figure supplements are available for figure 7:

**Figure supplement 1.** The time trace of nanodomains (*Figure 7h*) and the time-lapse of fluorescent images (*Figure 7*).

DOI: https://doi.org/10.7554/eLife.27744.032

**Figure supplement 2.** Another type of dynamics of the AMPA receptor domain in synapses.

*Figure 7 continued on next page*

*Figure 7 continued*

DOI: https://doi.org/10.7554/eLife.27744.033

**Figure supplement 3.** The last type of dynamics of the AMPA receptor domain in synapses.

DOI: https://doi.org/10.7554/eLife.27744.034

receptors are located. As we described above, currently the consensus viewpoint is that AMPARs enter PSDs from a highly mobile pool of AMPARs in the dendritic membrane, largely outside of the spine (*Figure 1a*, beginning with route 1). Other evidence indicates that the site of exocytosis is in the spine (route 3). In both cases, according to this data, a large mobile pool of extrasynaptic AMPARs originates from sites in the membrane and then diffuses into the PSD (route 2).

In this study, we find very little evidence for a sizeable pool of mobile extrasynaptic receptors in or around the spines. Most of the previous evidence in support of this AMPAR pool was from studies using super-resolution bQD-tagged AMPARs (*Opazo et al., 2012*; *Giannone et al., 2010*; *Nair et al., 2013*). Indeed, we too observe this behavior with bQDs (*Figure 3*). However, when using when AMPARs that are tagged by significantly smaller fluorophores—sQDs or organic fluorophores (*Figures 4* and *5*)—they are overwhelmingly in the synapse:~84–97% of AMPA receptors were immobilized within PSDs and only 5–16% were freely diffusing in the dendrite membrane. Consequently, we believe that bQDs are too large to enable AMPARs to enter the synaptic cleft. We therefore believe that the previous results with bQDs have been misleading.

Our results are consistent with studies using a variety of techniques that have found that AMPA receptors are largely localized to synapses and a relatively small extrasynaptic pool exists. First, electrophysiology has estimated the population of receptors in the synapses as well as in the dendrites. Cottrell *et al.* observed the density of AMPARs by measuring the glutamate-evoked response current through glutamate stimulating in synaptic regions or in dendrites. They observed the high current signal after glutamate stimulation in synaptic regions, but very low current signal with stimulation in extra-synaptic regions. Through comparing signal strength, they concluded that less than 1% of AMPA receptors were populated in the extra-synaptic regions after synaptogenesis by 11 days (or more) in vitro (DIV) (*Cottrell et al., 2000*). Second, studies using fluorescence microscopy showed that AMPARs (*MacGillavry et al., 2013*) were co-localized with PSD-95 in dissociated cultured neurons, indicating that they were heavily colocalized with the synaptic regions. In addition, it was reported that AMPA receptors were mostly localized at the synaptic region in hippocampal neurons (*Passafaro et al., 2001*). They observed that more than 75% of AMPA receptors, both subunits GluA1 and GluA2, were colocalized with Shank at the synaptic regions. Also, their spatial and temporal fluorescence images showed that the exocytosed receptors accumulated over time (~30 min) at the synaptic regions. Third, electron microscopy (EM) was used to show that AMPA receptors were localized at synapses (*Kharazia et al., 1996*), although they could also exist in extrasynaptic domains (*Tao-Cheng et al., 2011*), especially after synaptic excitation at exocytosis and/or endocytosis sites. All these observations indicate that most AMPARs are localized and clustered at synaptic PSDs in mature neurons.

The question arises as to why we observed so few highly-mobile extrasynaptic AMPARs. One possible reason is that in our protocol we have approximately 6 min of dead time with the addition of fluorophores and their binding to cell-surface AMPARs. It is likely many of the labeled AMPARs are diffusing into PSDs during the dead time and we do not observe them. Another possibility is that once AMPARs enter PSDs their residence time there is lengthy and we do not observe AMPARs exiting from PSDs because the rate of exit is much slower compared to the time period (usually 50 s) we use to monitor AMPAR dynamics.

We also performed experiments where we compared bQD-tagged AMPARs to sQD-tagged AMPARs with and without removing the ECM (*Figure 6*). In this case, only the extrasynaptic bQD-tagged AMPARs, which are largely sterically hindered from entering PSDs, were affected. Previous studies found that ECM was acting as a diffusion barrier for extrasynaptic AMPARs during their membrane diffusion from extrasynaptic sites to entry into PSDs (*Frischknecht and Gundelfinger, 2012*). Our data is consistent with this conclusion and also supportive of our findings that show the different localization of bQD-tagged AMPARs and sQD-tagged AMPARs. The diffusion of sQD-tagged AMPARs was not affected by enzymatic removal of the ECM because they are almost all within the PSDs away from the effects of ECM removal. Diffusion of bQD-tagged AMPARs was significantly

affected because they are largely highly mobile and extrasynaptic. Our findings on the bQD-tagged AMPARs are almost identical to the previous findings of others looking at the effects of ECM removal on bQD-tagged AMPARs (*Frischknecht et al., 2009*).

We conclude that the vast majority of AMPARs rapidly enter PSDs and are confined to sub-synaptic diffusion in nanodomains. The diffusion constant is about 10x slower than that of free extrasynaptic AMPARs. The AMPARs therefore do not accumulate in extrasynaptic membranes as previously inferred in studies using AMPARs labeled with bQDs. In previous studies, the findings that AMPARs accumulate in a highly mobile pool of AMPARs in extra-synaptic dendritic membranes suggested that this AMPAR pool existed because of limits in the numbers of AMPARs that are stabilized within PSDs (*Opazo et al., 2012*). According to Opazo et al., AMPARs are not stabilized within PSDs until 'slots', defined as PSD place-holders for AMPARs become free inside the PSD. Consequently, in this model, AMPARs build up outside of PSDs. We do not observe the highly mobile AMPAR pool when bQDs are replaced by a variety of smaller (organic and sQDs) fluorophores and linkers indicating that the highly mobile pool is an artifact of using bQDs attached to the AMPARs. Thus, the number of synaptic AMPARs does not appear to be limited by whether PSD slots are already occupied.

The picture that emerges from our data is that after entry into PSDs, AMPARs are mobile but constrained within confined sub-synaptic nanodomains that are surprisingly stable, lasting for more than 15 min. It is likely that the AMPAR entry *into* PSDs is more rapid than the membrane diffusion rate *to* the PSDs. If this were not case, there would be accumulation of a mobile extra-synaptic pool, similar to what is observed when bQDs are attached to the AMPARs. The positioning of these nanodomains is likely critical for the AMPARs to receive glutamate from the presynaptic release sites (*Tang et al., 2016*; *Jeyifous et al., 2016*). Our conclusion that the number of synaptic AMPARs is not limited by whether PSD slots are occupied suggests that other factors regulate AMPAR numbers in PSDs. Potential candidates for what limits AMPAR numbers in PSDs are AMPAR exocytosis rate (*Figure 1a*, route 1), AMPAR endocytosis rate (*Figure 1a*, route 3) and/or barriers such as extracellular matrix (*Figure 6*) that regulate AMPAR membrane diffusion on the membrane (*Figure 1a*, route 2).

## Materials and methods

### Microscope

Experiments were performed with a Nikon Ti Eclipse microscope with a Nikon APO 100 X objective (N.A. 1.49). The microscope stabilizes the sample in z-axis with the Perfect Focus System. An Agilent laser system MLC400B with four fiber-coupled lasers (405 nm, 488 nm, 561 nm and 640 nm) was used for illumination. Elements software from Nikon was used for data acquisition. A back illuminated EMCCD (Andor DU897) was used for recording. For 3-D imaging, a cylindrical lens (CVI Melles Griot, SCX-25.4-5000.0- C-425-675) of 10 m focal length was inserted below the back aperture of the objective. A motorized stage from ASI with a Piezo top plate (ASI PZ-2000FT) was used for x-y-z position control. A quad-band dichroic (Chroma, ZT405-488-561-640RPC) was used and band-pass emission filter 525/50, 600/50, 710/40, 641/75 was used for fluorescence imaging.

### Stage drift correction

Stage drift is a significant problem in taking super-resolution images. As we reported earlier (*Lee et al., 2012*), making fiduciary-marked patterns on coverslips is a very efficient method to correct stage drift instead of using the previously favored fluorescent beads and gold nano-particles as fiduciary markers to correct stage drift. The perfectly stable fiduciary-marked patterns proved far superior to correct stage drift since it is impossible to perfectly immobilize beads or nano-particles on a coverslip surface, making it especially difficult to image live cells. More importantly, randomly spreading beads or nano-particles is a critical drawback for using them as fiduciary markers because it needs optimal number of fiduciary markers in the region of interest, whereas our fiduciary markers are a uniform pattern. Thus, the fluorescent beads and gold nanoparticles that have typically been used to correct stage drift are not effective, especially when imaging live cells.

We have previously reported that mammalian cells can be stably cultured on fiduciary-marked coverslips without any other coating or treatment; however, although we had no problems taking

super-resolution images of mammalian cells with this fiduciary pattern, we were not able to culture dissociated neurons on fiduciary-patterned coverslips.

In general, poly-(L/D)-lysine coating is necessary to culture dissociated neurons on glass coverslips because the cell membranes of neurons are negatively charged. Poly-lysine coating is not sufficient for neurons to adhere the fiduciary marker coverslips. We then employed laminin to coat coverslips. As shown in *Figure 2—figure supplement 3(a)*, we first coated poly-lysine after treating coverslips in oxygen plasma and then applied an additional coating of laminin on the fiduciary-patterned coverslips. We found that dissociated neurons could be stably cultured on fiduciary-patterned coverslips after coating them with both poly-lysine and laminin, sequentially, as in *Figure 2—figure supplement 3(b)*. In terms of connection between neurons, we could obtain comparably good neurons on the fiduciary marked coverslip as we could culture them on the poly-lysine coated coverslip. More details to coat poly-lysine and laminin on the fiduciary marked coverslips were described in the method section.

As shown in *Figure 2—figure supplement 3(b–d)*, once neurons have been stably cultured on fiduciary-patterned coverslips, they can then be successfully transfected. Furthermore, IR illumination, as shown in *Figure 2—figure supplement 3(b)*, is effective to allow clear observation of uniformly patterned fiduciary markers without affecting the visibility of fluorescence color channels. Hence, these coated fiduciary patterned coverslips are very useful to exclude stage drift, allowing for the culture, transfection, and super-resolution imaging and tracking of live neurons.

Here, to demonstrate the method for taking super-resolution fluorescence images, neurons were transfected with two plasmids, one allowing expression of the post-synaptic protein homer1 with a photoactivable fluorescence protein, mGeos or mEOS3.2, and the other allowing expression of glutamate receptors, AMPARs with AP tag to biotinylate. We then labeled the glutamate receptors using qdots conjugated with either streptavidin or anti-GFP nanobodies. As shown in *Figure 2—figure supplement 3(c)*, at 530 nm ~ 700 nm of emission in wavelength, we observed fluorescence from the proteins and receptors without any auto-fluorescence from the fiduciary pattern. We could clearly observe the fiduciary pattern in the IR channel and the live cells in the visible channels, as shown in *Figure 2—figure supplement 3(d)*. Fiduciary markers in the IR image are tracked, fitting the fiduciary markers using either the Gaussian or the Airy function. The IR camera electronically synchronizes with the visible camera so that the traces of fiduciary markers as recorded by the IR camera accurately calculates the stage drift at the time the cells are imaged. We can then correct the super-resolution cell images for stage drift accordingly.

The fiduciary marker trace corresponds to the stage drift. Thus, through this post-processing, we can correct for stage drift. Since fiduciary markers are stationary, stage drift can be much more accurately calculated using them than when using more conventional existing methods of calculating stage drift with fluorescent beads or gold nano-particles. As shown in *Figure 6*, using fiduciary markers, stage drift can be controlled within an error of around 1 nm for x and y and 15 nm for z.

## Chromatic aberration

Another technical difficulty is chromatic aberration. The three-dimensional multi-color super-resolution imaging technique allows us to observe the interaction between two different species/types of proteins or organelles in cells; however, the color shifts that occur in different color channels is a concern to take images can be even more pronounced and problematic in super resolution images different color channels. These color shifts will cause the images to be slightly different. Even though we observe exactly the same object at different color channels, the images will be a little mismatched in different color channels. Since the refractive index of light depends on its wavelength, this chromatic aberration causes focus on an object to be detected in a little different place in each color channel. This means that we have to correct this difference chromatic aberration for studying the interaction between different proteins or organelles when using multi-color super-resolution imaging.

Most commercially available objective lenses are designed to optically minimize chromatic aberration. Based on our measurements, this optical design for objective lenses will corrects within the margin of error of about 100 nm; however, given that the spatial resolution of super-resolution microscopy is only about 20 ~ 30 nm, a margin of the error of 100 nm is a significant error. This means that achromatic optical design is not sufficient to correct chromatic aberration, especially in super-resolution imaging.

Using our method, chromatic aberration can be corrected within a margin of error of 2 nm. We accomplish this with as reported (*DeWitt et al., 2012*; *Pertsinidis et al., 2013*; *Wang et al., 2014*). In order to correct chromatic aberration, we made a nano-hole pattern on a silver- coated coverslip. Using a thermal evaporator, a glass coverslip is coated with silver at a thickness of about 100 nm using a thermal evaporator. The nanohole pattern is produced by fast ion beam (FIB) equipment, sputtering holes of about 100 nm in size every 1.5 μm. Holes are about 100 nm in diameter and are located every 1.5 μm.

We took images for the nanoholes on three at the different color channels. As shown in supporting figure *Figure 2—figure supplement 7(a–d)*, the positions of the nanohole patterns are slightly different on each color channel. In *Figure 2—figure supplement 7(d)*, the red and green spots are the nanoholes in the green channel and in one of two red channels, shown by zooming in on the image of the the overlaid two color images. The significantly different position of the two sets of nanohole patterns is a result of shows the positions of red and green spots are significantly different, which is caused by chromatic aberration.

We measured the chromatic aberration between the green (GFP) 530 nm channel and red (RFP) 600 nm channel and 705 nm channels. As shown in *Figure 2—figure supplement 7(e)*, the average difference in the location of the nanoholes in between two the nanohole pattern on channels 530 nm and 600 nm is about 80 nanometers. We found a similar range for the difference between nanohole locations in any two of the three-color channels. This means that, when obtaining multi-color super-resolution images, the error of co-localizations of any two colors is expected possible to be at least approximately 80 nm.

Furthermore, we also evaluated the validity of the mapping functions for 3D imaging. We obtained the mapping function at a focal point of z = 0, focal point, with an error of 2 nm, and, as shown in *Figure 2—figure supplement 7(e)*, this mapping function can be applied is valid up to within z= ±400 nm with an error of only 4 nm. when the error is 2 nm at z = 0. Actually, we obtained the mapping function at z=200 nm, 400 nm, 600 nm, and 800 nm and, as represented in *Figure 2—figure supplement 7(e)*, the error is a little smaller than when using the mapping function obtained at z = 0; however, since a 3 D super-resolution image using astigmatism is not discrete along the z axis, it is not easy to have mapping functions along the z axis. At focal points above 400 nm, the error dramatically increased. We obtained the mapping function at z = 200 nm, 400 nm, 600 nm, and 800 nm, as represented in *Figure 2—figure supplement 7(e)*. At focal points above 400 nm, the error dramatically increased. Although the error is a little smaller at focal points higher on the z axis than when using the mapping function obtained at z=0, we conclude that when getting 3D images using astigmatism, the mapping function at z=0 is accurate enough to correct chromatic aberration. This is because a 3 D super-resolution image using astigmatism is not discrete along the z axis. Hence, it would be too time consuming to try to obtain more accurate mapping functions all along the z axis.

## Cell culture and labeling

Primary hippocampal cultures were prepared from E18 rats according to UIUC guidelines as previous described (*Cai et al., 2014*) with the following modifications. In brief, hippocampal tissues were dissociated in 3 mg/mL protease and plated on 25 mm coverslips coated with 1 mg/mL poly-l-lysine and laminin. Neurons were cultured at 37°C with 5% $CO_2$ in neurobasal media (ThermoFisher, 21103049) with B-27 supplement (GIBCO), 2 μM glutmax and 50 unit/mL penicillin and 50 unit/ml streptomycin. On 12–13 days in vitro (DIV), neurons were co-transfected with Homer1c-mGeos (1 μg/coverslip) with GluA2-AP (1 μg/coverslip) and BirA-ER (1 μg/coverslip), and Homer1c-mGeos (1 μg/coverslip) for AMPA receptors by using Lipofectamine 2000 transfection reagent. At 24 ~ 72 hr after transfection, the coverslips were transferred to warm imaging buffer (HBSS supplemented with 10 mM Hepes, 1 mM MgCl2, 1.2 mM $CaCl_2$ and 2 mM D-glucose) for 5 min incubation and mounted onto an imaging dish (Warner RC-40LP). In the imaging dish, neurons were incubated in imaging buffer containing QDs and casein (~400 times dilution for bQDs, and ~80 times dilution for sQDs; stock solution purchased from Vector labs, SP-5020) for 5 min at 30°C and washed with 10 ml of imaging buffer. Finally, 3 mL of imaging buffer was added to the imaging dish that was subsequently mounted on the microscope. For ECM removal, we enzymatically removed the ECM using hyaluronidase (HX0514 – Calbiochem, 100 units/ml left overnight at 37° with 5% $CO_2$). For labeling endogenous AMPARs using antibody, we conjugated Alexa647 NHS ester (Thermo-Fisher:

A37573, Waltham, MA) with anti-GluR2(Anti-Glutamate Receptor 2 Antibody, extracellular, clone 6C4: Sigma-Aldrich: MAB397, St. Louis, MO)

## Expression and purification of mutant mSA

To express mSA, *E. coli* strain BL21(DE3) pLysS was transformed with an appropriate expression vector and plated on Luria Bertani (LB) agar containing ampicillin. Several colonies were selected from the plate for overnight growth at 37°C in an LB starter culture containing 200 µg/ml ampicillin. The following morning, the starter culture was diluted 100 fold into Terrific Broth (TB) containing 0.05% glucose, 0.5% glycerol, 0.2% α-lactose, 1 mM MgSO4, 900 µM biotin, and 200 µg/ml ampicillin. The cells were grown at 37°C and 300 rpm until $OD_{600}$ reached 0.3, at which point the shaker temperature was reduced to 20°C. Once the culture has reached $OD_{600}$=0.6, IPTG was added to the final concentration of 75 µM and the shaker speed was increased to 375 rpm. After 24 hr induction at 20°C, cells were removed by centrifugation and the culture medium containing secreted mSA was collected for purification. The medium was sonicated with a 200 W sonicator (30 s on, 30 s off for 4 cycles at 50% amplitude capacity). The pH of the culture medium was adjusted to 7.5 using NaOH and imidazole was added to 10 mM. The medium was then centrifuged at 12,000 rpm for 20 min to remove any precipitates. The clarified medium was passed through a column packed with Ni-NTA Superflow Agarose pre-equilibrated in PBS and 10 mM imidazole. The column was then washed with PBS and 20 mM imidazole. Finally, the bound protein was eluted with 300 mM imidazole in PBS. The eluted protein was concentrated and buffer exchanged to 100 mM glycine buffer pH 2.3 using a centrifugal filter to remove bound biotin. Finally, the sample was buffer exchanged to PBS. Yields were estimated based on $A_{280}$ measurements and purity was assessed by SDS-PAGE with Coomassie staining.

## Characterization of mutant mSA

All streptavidin homologs have a conserved three dimensional structure, but differences in the binding pocket residues lead to differences in biotin binding. For example, the main chain root mean squared deviation between mSA and streptavidin is 0.82 Å but there is ~$10^5$ fold difference in their binding affinities. Although extremely high affinity is not needed for labeling live cells, rapid dissociation of bound biotin interferes with detection and measurement because it results in loss of fluorescence intensity over time. As such, improving the dissociation kinetics (i.e. slowing the off rate $k_{off}$) of mSA is important to develop a useful imaging reagent. We have previously demonstrated that replacing S25 with histidine reduces $k_{off}$ by 7.5 fold by blocking solvent entry into the binding pocket through steric hindrance (*Demonte et al., 2013*). Keeping water molecules out of the binding pocket is important because they compete with mSA-biotin hydrogen bonds and lead to biotin dissociation. S25 is located in the loop between β−strands 1 and 2 (L1,2), which is found at the subunit interface of streptavidin but is solvent exposed in mSA. Therefore, blocking solvent entry in mSA in part involves recreating the lost physical barrier near L1,2.

To improve the dissociation kinetics further, we modeled the binding pocket of mSA using high affinity dimer bradavidin II (Brad2), which binds biotin with $K_d$ <$10^{-10}$ M (*Helppolainen et al., 2008*). L1,2 and the binding loop L3,4 adopt nearly identical conformations in mSA and Brad2, thus allowing key residues to be reliably modeled in mSA. The Brad2 residue corresponding to mSA S25 is R17, which makes a hydrogen bond and a salt bridge with G103 and D104, respectively, to form a solvent barrier around L1,2 as shown in *Figure 4—figure supplement 1(a, b)*. Arginine is larger than serine and histidine and is therefore more effective in blocking solvent entry. To reconstruct the solvent barrier in mSA, we mutated S25 to R, and measured the dissociation rate of the mutant mSA using fluorescence polarization spectroscopy. The dissociation rate of mSA-S25R is slower than that of mSA or mSA-S25H.

Brad2 also contains F42 in L3,4 to create a hydrophobic lid over bound biotin and trap the molecule in the binding pocket. Mutating F42 to alanine significantly reduces the affinity of interaction, indicating that the hydrophobic lid also plays an important role in biotin binding. We therefore introduced the corresponding mutation T48F in mSA-S25R to create a hydrophobic binding pocket and further reduce the dissociation rate. The measured $k_{off}$ of mSA-S25R/T48F (mSA-RF) is 37 fold lower compared to original mSA, that is. $t_{1/2}$=402 min, and ~5 fold lower than the best mutant to date, thst is mSA-S25H (*Figure 4—figure supplement 1c*). The single point mutant mSA-T48F resulted in

only limited improvement in biotin dissociation ($t_{1/2}$=13 min), indicating that the two mutations function cooperatively. We also tested an alternative design of the hydrophobic lid with a tryptophan, mSA-S25R/T48W/D124T (or mSA-RWT), which had similarly slow dissociation kinetics, $t_{1/2}$=330 min (*Figure 4—figure supplement 1c*). Therefore, construction of solvent barriers through cooperative mutations in L1,2 and L3,4 can modulate the dissociation kinetics and improve the mSA-biotin interaction.

To demonstrate the use of new engineered mSA in cell labeling, we expressed a biotin acceptor peptide (AP) fused to CFP and a transmembrane helix (TM) on the surface of HEK293. We then added purified *E. coli* biotin ligase, BirA, to the cell culture to induce site-specific biotinylation of AP. The cells were fixed and labeled with mSA-RF—EGFP for imaging by confocal microscopy. The CFP positive cells were selectively labeled, suggesting high specificity of interaction between mSA-RF and biotin (*Figure 4—figure supplement 1d and e*). Furthermore, the intensity of EGFP fluorescence remained high after 1 hr, indicating that the interaction is stable and mSA-RF remained stably bound during measurement.

In order to verify whether SA-qdots were in fact cross-linking, we compared the diffusion coefficient of mSA-qdots and SA-qdots on the lipid bilayer at different concentrations (0.2% and 1.0%) of biotinylated lipid molecules. If a qdot binds to one biotinylated lipid molecule, the resulting labeled lipid molecule will diffuse at the same rate as a free diffusion of a lipid molecule; whereas, if a qdot binds to multiple biotinylated lipid molecules, the diffusion rate will significantly slow down as shown in *Figure 4—figure supplement 2* .

We first determined whether there is any difference in the diffusion rate of molecules labeled in three ways: (1) those labeled with normal SA-qdot conjugates formed with big qdots (one conjugate has many streptavidin and numerous binding site on SA-big qdots); (2) those labeled with normal SA-qdot conjugates formed with small qdots (one conjugate has four binding sites from the streptavidin); and (3) those labeled with mSA-qdot conjugates using monomeric streptavidin and small qdots (one conjugate has only one binding site). We hypothesized that normal streptavidin conjugates with both small and big qdots would be much more likely to bind multiple biotins than would be the case with mSA conjugates. We found this to be the case, as represented in *Figure 4—figure supplement 3*. *Figure 4—figure supplement 3* shows that there was little difference in the diffusion rate of molecules labeled with normal SA-qdots, whether conjugated with small or big qdots: there were a large number of slow diffusive qdots for both normal streptavidin conjugates, indicating that they had cross-linked to multiple molecules, even at the 0.2% biotin concentration shown in *Figure 4—figure supplement 3*. At 1.0% biotin, there was even a larger population of slow diffusive qdots, that is, cross-linked qdots. By contrast, in the case of mSA qdot conjugates, we found that most qdot conjugates were freely diffusive at 0.2% as well as at 1.0% biotin concentration; that is, mSA-qdots do not show a slow diffusive population of labeled molecules at either 0.2% or 1.0% concentrations. These observations showed that normal streptavidin frequently cross-links to multiple molecules while monomeric streptavidin did not.

In order to further examine cross-linking of normal and monomeric streptavidin, we compared the run length of kinesin (K432, biotin at C-terminus). It is well known that when the cargo (in this case the streptavidin-qdot conjugate) binds to multiple motor proteins, the cargo will be transported over a longer distance than when the cargo binds to a single motor protein. (*Beeg et al., 2008*; *Furuta et al., 2013*) A single motor protein transports cargo approximately 1 um, while multiple motor proteins can walk farther than 1 um. Thus, if we compare the run length of kinesin using SA-qdots and mSA-qdots at different ratios of kinesin and qdots, then we can examine the possibility that one qdot is binding to multiple proteins versus binding to a single protein. For accuracy of labeling, we want to confirm that binding is to a single protein and a shorter kinesin run length will indicate this.

In the case of mSA-qdots, we found the run length was the same as that of a single kinesin at a ratio of 2:1 (two kinesin molecules to one qdot). As expected (*Figure 4—figure supplement 4*), the run length was not changed even when the ratio of kinesin to qdots was increased to 6:1. This means that mSA-qdots do not bind to multiple motor proteins. By contrast, in the case of SA-qdots, the run length of kinesin was more than the typical run length of kinesin, as shown in *Figure 4—figure supplement 4*. Even at a ratio of 2:1, the histogram of run lengths shows that some kinesin walked longer than the typical kinesin run length. At 6:1, we found that the run length significantly

increased, indicating cross-linking. These results mirror the results we obtained in our first experiment about diffusion rates of labeled molecules in the lipid bilayer.

## Track AMPA receptors

After focusing the sample in bright field, the Perfect Focus System was activated to minimize the sample drift in z direction. The samples were then scanned in the GFP channel (488 excitation, 525/50 emission) to locate transfected cells. A fluorescent image of the cells was taken for reference. To track the QD labeled receptors, 488 nm or 561 nm lasers was used for excitation in the hi-low-fluorescence mode with an appropriate band-pass filter for collecting the fluorescence. In terms of localization precision of fluorophores, as shown in *Figure 2—figure supplement 8*,organic dyes were about 11 ~ 21 nm for x and y (Atto647N – 12 nm for both x and y) and CF633- 21 nm for x and 12 nm for y) and about 32 nm (Atto 647N)/42 nm (CF633) for z. Qdots were 9.6 nm for x, 8.2 nm for y, and 18 nm for z.

## Super-resolution imaging of post-synaptic density

After the tracking experiment, the PALM (*Betzig et al., 2006*) experiment was carried out on the same neurons. PALM was used for super-resolution of post-synaptic density. Post-synaptic protein Homer1c was used as the PSD marker and its C- terminus was fused to photoactivatable protein mGeos. A 100 ms 405 nm laser pulse was used to activate mGeos proteins from dark to green fluorescent state. The sample was then excited with a 488 nm read out laser and emission was collected with a 535/50 band-pass filter. The cycle was repeated for 200–300 times, collecting 4000–6000 frames. The z calibration was created following method described by Huang *et al.* (*Huang et al., 2008*) using fluorescent beads on a glass surface and applied to both tracking and PALM data.

## STORM imaging for AMPA receptors

Cy3B-STORM was done following the procedure described on Joshua et al (*Vaughan et al., 2012*). We labeled neurons with Cy3B-SA (2 µM) following the procedure of sQD labeling. We then fixed the cells with a solution of 4% PFA and 0.1% Glutaraldehyde in HBSS for 10 min. Wash with three tines PBS. Treat the sample with 10 mM of NaBH4 in PBS freshly prepared for 10 min. Wash with PBS. Before imaging, we add imaging buffer consisting of 5 µL of PCD, 20 µL of PCA, 4 µL of Trolox in 471 µL of T-100 (100 mM Tris at pH 8.0). Images were acquired at 10 Hz using 405 nm laser at low power to do activation and 561 nm laser to excite fluorescence.

In order to image native AMPARs, we labeled AMPARs using Anti-GluA2-Alexa647 after fixation. For STORM imaging, we added imaging buffer consisting of 5 mM MEA (Sigma: 30070, St. Louis, MO) solution (~pH 8.0) and additionally added 40 mM Sodium D/L-lactate (Sigma: 71720, St. Louis, MO) and EC-Oxyrase (Sigma: SAE0010, St. Louis, MO) in PBS in order to improve the photostability.

## Data analysis

For the tracking data, centroids of the all the QDs were localized in all the frames and a map of all the places QDs visited were obtained. A Matlab code was used to recover the trajectories of the QDs. In brief, the code finds locations of QDs in time t, and searches for nearby QDs in time t+1 as the next point on the trajectory. In the 3-D single particle tracking experiment, the maximum displacement of a QD in one time step is set to be 1 µm. The trajectory range was obtained by calculating the range of the trajectory in the x-, y-, and z- direction, and using the maximum of the three parameters. The diffusion coefficients from the trajectories were calculated in Matlab by fitting the first 4 points of mean-square-displacement curve. For the PALM data, positions of proteins detected in each frame are localized, cluster analysis was used for identify synapses. To determine if a trajectory was synaptic, the centers of synapses were determined, and trajectories within 2 µm radius of each synapse were identified. For each of these trajectories, the average distance was calculated between the center of the nearest synapse and all points on the trajectory. Any trajectories with an average distance smaller than 0.55 µm were considered synaptic.

The visualizations of post-synaptic densities and sQD labeled AMPA/NMDA receptor tracking trajectories were created using VMD 1.9.2 (*Humphrey et al., 1996*), a widely-used program for visualizing and analyzing macromolecular structures and molecular dynamics (MD) simulation trajectories.

VMD is optimized for dealing with large data sets containing millions of particles and thousands of trajectory frames, as encountered in the current study.

## Acknowledgements

This work was partially supported by NIH NS090903 and NIH NS100019 to PRS and WNG and by NSF PHY-1430124 to PRS and by NSF CBET-1264051 to SP.

## Additional information

### Funding

| Funder | Grant reference number | Author |
| --- | --- | --- |
| National Institutes of Health | NIH NS090903 | William N Green<br>Paul R Selvin |
| National Science Foundation | PHY-1430124 | Paul R Selvin |
| National Science Foundation | CBET-1264051 | Sheldon Park |
| National Institutes of Health | NIH NS100019 | William N Green<br>Paul R Selvin |

The funders had no role in study design, data collection and interpretation, or the decision to submit the work for publication.

### Author contributions

Sang Hak Lee, Conceptualization, Data curation, Formal analysis, Investigation, Visualization, Methodology, Writing—original draft, Writing—review and editing; Chaoyi Jin, Data curation, Investigation, Visualization, Methodology; En Cai, Data curation, Software, Visualization, Methodology; Pinghua Ge, Investigation, Methodology; Yuji Ishitsuka, Duncan Nall, Methodology; Kai Wen Teng, Data curation, Formal analysis, Methodology; Andre A de Thomaz, Data curation; Murat Baday, Data curation, Methodology; Okunola Jeyifous, Daniel Demonte, Christopher M Dundas, Resources, Methodology; Sheldon Park, Resources, Methodology, Writing—review and editing; Jary Y Delgado, Writing—review and editing; William N Green, Conceptualization, Investigation, Writing—original draft, Writing—review and editing; Paul R Selvin, Conceptualization, Resources, Supervision, Funding acquisition, Investigation, Writing—original draft, Project administration, Writing—review and editing

### Author ORCIDs

Sang Hak Lee (iD) http://orcid.org/0000-0003-3434-076X
William N Green (iD) http://orcid.org/0000-0003-2167-1391
Paul R Selvin (iD) http://orcid.org/0000-0002-3658-4218

### Ethics

Animal experimentation: Primary hippocampal cultures were prepared from E18 rats according to UIUC guidelines. All rats were handled according to approved institutional animal care and use committee (IACUC) protocols (#15254) of UIUC.

### Decision letter and Author response

Decision letter https://doi.org/10.7554/eLife.27744.035
Author response https://doi.org/10.7554/eLife.27744.036

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
