## [Decision Letter]

[Editors’ note: a previous version of this study was rejected after peer review, but the authors submitted for reconsideration. The first decision letter after peer review is shown below.]

Thank you for submitting your work entitled "Super-resolution Imaging of Synaptic and Extra-synaptic Pools of AMPA Receptors with Different-sized Fluorescent Probes" for consideration by *eLife*. Your article has been reviewed by three peer reviewers, one of whom is a member of our Board of Reviewing Editors, and the evaluation has been overseen by a Senior Editor.

Our decision has been reached after consultation between the reviewers. Based on these discussions and the individual reviews below, we regret to inform you that your work will not be considered further for publication in *eLife*. There were major concerns regarding the utility of the small quantum dots as compared to organic dye labeling methods and regarding the neuronal culture experiments as detailed below.

Reviewer #1:

This work suggests that AMPA receptors are primarily localized to individual to synapses rather than diffusing in the membrane of the dendrite outside synapses.

Major concerns:

1) Expression level of GluA2. It has been shown that over-expression of GluA2 (GluR2) increases spine size and alters morphology (doi:10.1038/nature01781 albeit not acutely, as performed here). How does expression look when using an antibody targeting all AMPA subunits in fixed tissue that is or is not expressing tagged-GluA2? A citation would probably be sufficient absent data.

2) Are multiple release sites ever observed in non-fixed cells using any of the probes (for example, compare Figure 2)? How certain are the authors of the location of the plasma membrane? The fixation solution they use permeabilizes the membrane, is it possible these multiple release sites are internal aggregations of tagged-AMPA receptors? Or do they represent bonafide release sites?

Reviewer #2:

This paper investigates the role of label size in controlling the distribution and dynamics of tagged surface receptors, here in particular the AMPARs. Position and dynamics of these receptors are critical in many aspects of neuroscience. The general theme that antibody-labeled receptors may not enter synapses has been present in the field for many years, but a thorough and systematic examination of the problem has not been published. A few comparisons among probes have been published, typically showing at least a mild size-dependence of synaptic diffusion coefficient, and concluding that small tracking tags are to be preferred. Thus, there is ample value in continued searching for the ideal probes.

The particular value of quantum dots may warrant special treatment, since they have been revolutionary in driving discovery and characterization of protein motion in and on cells. However, their predominance in single-particle tracking studies recently has been substantially lessened with the advent of brighter, more stable dyes and the proliferation of alternative labeling methods. For instance, QD photostability underlies their prime utility, but this paper like others obtains high-resolution tracks based on Atto dyes. The ~10 nm precision achieved in typical experiments here is only slightly better than for the best dyes; much of that error stems inevitably from protein motion and other aspects of the imaging strategy not from error that could be overcome with more signal photons. Thus, the impact of the paper is undermined by the fact that the authors present no particularly compelling reason to use QDs over other label types.

Fundamentally, the data in the paper are weak in that it appears that most results were obtained from single example neurons. For instance, the 2566 trajectories from the one bQD neuron in Figure 6 are by no means independent measurements.

Related, the idea that a change in diffusion from 1 to 3 days transfection is at best indirect evidence of cross-linking. Won't the difference in α vs β peak height difference be extremely sensitive to the proportion of the imaged area of the dendrite occupied by synapses? Isn't a difference thus expected during this developmental period of intense synaptogenesis? Because this varies widely between cells, this can only be addressed more systematically than is done here.

If the major point is to compare large and small QDs, can't the two probes be applied together and imaged in the same neurons at the same time? This would give much higher confidence in any differences observed.

Reliance on overexpressed AMPARs undermines the potential relevance of the conclusion. We do not know how the expressed receptors recapitulate, avoid, or interfere with the expression pattern of endogenous receptors.

Problems associated with the size of the quantum dot and the ability of a multi-liganded label to oligomerize should not be conflated. Small, polyvalent labels can cross-link receptors yet presumably can enter the synapse well.

The paper is riddled with grammatical and language errors and is far from suitable for publication. It will require extensive editing before any new submission.

Reviewer #3:

The purpose of this study is to determine whether AMPARs are delivered to the synapse by lateral diffusion or exocytosis. This study uses extracellularly tagged GluA2 to track the movement of AMPARs and determine their ability to localize at synapses. They use different sized tags to determine what effect this has on synaptic trafficking. Not surprisingly they find that large QDs impede synaptic trafficking more than small QDs. It is unclear to me how the authors use this data to argue that synaptic AMPARs are delivered by exocytosis. The logic escapes me.

There has been a lot of work using QDs carried out by the Choquet lab. The authors fail to discuss this work adequately. Do they agree with this work or not. The author's cite a few old papers to support their conclusion that there are few nonsynaptic AMPARs. However, there are many papers that unequvically show the existence of endogenous AMPARs on extrasynaptic mambranes. For instance, there is EMgold (e.g., Tao-Cheng et al., 2011), outside out dendritic patches (e.g., Andrasfalvy et al., J. Physiol. 552: 35, 2003) and 2P glutamate uncaging on dendritic shafts (e.g., Soares et al., J. Neurosci. 2013).

[Editors’ note: what now follows is the decision letter after the authors submitted for further consideration.]

Thank you for resubmitting your work entitled "Super-resolution Imaging of Synaptic and Extra-synaptic Pools of AMPA Receptors with Different-sized Fluorescent Probes" for further consideration at *eLife*. Your revised article has been favorably evaluated by Richard Aldrich as Senior editor and three reviewers, one of whom is a member of our Board of Reviewing Editors.

The manuscript has been improved but there are some remaining issues that need to be addressed before a final decision can be made, as outlined below:

1) Overall, the presentation is rather impenetrable. The key comparisons are spread over the paper, presented confusingly, and are diluted by unrelated details. A tendency to state important results as true without quantitative support is still also a weakness here. A serious effort should be made to improve the flow of the paper and presentation of the data and conclusions.

2) Data was added in Figure 5, but 5D vs 5E are so obviously different that it undermines the conclusion that they are the same in the way the authors stress. If these images were analyzed in the same way, how could one distribution cut off abruptly at ~1 μm where the other extends smoothly to 2 um? Plus, there's obviously a prominent peak at ~0.4 μm in 5D that is absent in 5E. Isn't this critical to interpreting the distribution of receptors or the performance of the labeling strategies? Sure, the general point holds and supports the general conclusion, but to baldly state that they are the same when they in fact are quite different really suggests that the authors haven't evaluated this question very critically.

3) Figure 7 was added, which is interesting and a good use of the QDs. The volume illustrations in panels C, D, E are nearly uninterpretable, though, since the overlays become almost black. Is there evidence that the receptor imaged in different epochs really is the same receptor? Surely there must be some exchange, increases, or decreases in number over 15 min. More seriously, there is little attempt to use the approach to evaluate some of the key questions that it raises, and it boils down mostly to a couple of anecdotes rather than a serious contribution.

4) Figure 7 also provides a new but confusing operational definition of a nanodomain (paragraph three in subsection “Using sQDs for longer measurements of AMPARs in PSDs”). A nanodomain in Nair and MacGillavry is a delineated subsynaptic region where receptors are denser than elsewhere in the synapse. Through most of this paper, "nanodomain" is used pretty much synonymously with "synapse" since there's no attempt to distinguish subregions of the synapse and no effort at quantifying high or low areas of receptor density. Indeed, the size of the "nanodomain" in 7F is probably close to or equal to the entire synapse. Then in 7 to suggest that where one receptor is present for long periods is the same as one of those other nanodomains is confusing and I believe premature. I think it would be of interest to determine whether further analysis of the sQD tracking can help relate these terms to one another, but that surely would be many experiments and is well beyond the scope of this paper; instead, please avoid the confusion entirely and not use "nanodomain" so liberally particularly in this section of the paper.

5) There is no definition of "constrained" diffusion in the paper, either mathematically or otherwise. Surely slow is not the same thing as constrained, as implied in subsection “AMPAR-labeling: Diffusion constants and trajectory range”? Don't the authors want to distinguish between "confined" and slow, and to measure the confinement radius? Or is "constrained" just defined as being "pretty slow and pretty short range"?

6) The first conclusion in the Discussion (receptor distribution around spines) is not even dealt with directly in the paper; no effort is made to measure density on spines, and no number is even given for the total proportion of receptors outside synapses-only the proportion of those that are outside the synapse but within 2 um.

7) There is an extensive literature using FRAP and optical antagonists to deduce receptor mobility, and this should be evaluated in light of the current results as well. FRAP results have shown mobile fractions ranging from 10 to 90% and have been key support for the idea that receptors continually exchange and diffuse into the synapse during LTP.

8) The final paragraph of the Discussion contains many premature statements.

– The PSD slot hypothesis long predates Opazo, Sainlos and Choquet, 2012. It stemmed from electrophysiological observations following molecular manipulations, not from imaging results.

– The slot hypothesis even as stated in Opazo does not predict that AMPARs cannot enter the PSD-it merely states that there is no way for synapses to anchor and retain them if they do enter.

– It is unclear why a small pool of extrasynaptic receptors suggests that there are or are not slots.

– The presence of clustered receptors in the synapse also does not seem to bear on whether a saturable molecular complex is involved in retaining them.

– If the authors wish to argue that mobility itself is a sign that slots do not exist, they need to make that argument more carefully and thoroughly. Isn't a receptor bound to almost any component of the PSD likely to be nearly completely immobilized?

– What's the difference between a slot and a place within the synapse where a receptor is constrained from leaving?

9) A comment from the original second reviewer seems to have been somewhat poorly answered: "Fundamentally, the data in the paper are weak in that it appears that most results were obtained from single example neurons. For instance, the 2566 trajectories from the one bQD neuron in Figure 6 are by no means independent measurements.". The authors reply that "The data is from independent measurements made from a minimum of 6 different areas per coverslip and 4 independent culture preparations. So the 2566 trajectories mentioned, in fact, are truly independent measurements." This reply is not fully correct. The 2566 trajectories are not truly independent from each other – surely at least some trajectories come from the same series of images (movies). The authors should make more efforts to point out how many trajectories were analyzed per movie, and how many movies they analyzed, from how many coverslips, and from how many independent cultures.

---

## [Author Response]

[Editors’ note: the author responses to the first round of peer review follow.]

Reviewer #1:This work suggests that AMPA receptors are primarily localized to individual to synapses rather than diffusing in the membrane of the dendrite outside synapses.Major concerns:1) Expression level of GluA2. It has been shown that over-expression of GluA2 (GluR2) increases spine size and alters morphology (doi:10.1038/nature01781 albeit not acutely, as performed here). How does expression look when using an antibody targeting all AMPA subunits in fixed tissue that is or is not expressing tagged-GluA2? A citation would probably be sufficient absent data.

We agree that this is a worthwhile experiment. Data was added which are for native-AMPA receptors (GluA2). We observed similar results to that done with transfection.

2) Are multiple release sites ever observed in non-fixed cells using any of the probes (for example, compare Figure 2)? How certain are the authors of the location of the plasma membrane? The fixation solution they use permeabilizes the membrane, is it possible these multiple release sites are internal aggregations of tagged-AMPA receptors? Or do they represent bonafide release sites?

This is an interesting question, however, is beyond the scope of this paper. In general, we did not permeabilize cells after fixation, so it was not possible to label internal tagged-AMPA receptors. Clearly the probe is not entering the cell cytoplasm under these conditions. For the experiments where the cells are fixed, protocols are used that show evidence of cell permeabilization. Furthermore, the staining with live and fixed cells is similar providing further evidence that with fixation we are not labelling intracellular aggregates of AMPARs. In terms of the position of plasma membrane, we are not certain the position of the plasma membrane, but we just make the boundary of the cluster of homer1 as the position of plasma membrane.

Reviewer #2:This paper investigates the role of label size in controlling the distribution and dynamics of tagged surface receptors, here in particular the AMPARs. Position and dynamics of these receptors are critical in many aspects of neuroscience. The general theme that antibody-labeled receptors may not enter synapses has been present in the field for many years, but a thorough and systematic examination of the problem has not been published. A few comparisons among probes have been published, typically showing at least a mild size-dependence of synaptic diffusion coefficient, and concluding that small tracking tags are to be preferred. Thus, there is ample value in continued searching for the ideal probes.The particular value of quantum dots may warrant special treatment, since they have been revolutionary in driving discovery and characterization of protein motion in and on cells. However, their predominance in single-particle tracking studies recently has been substantially lessened with the advent of brighter, more stable dyes and the proliferation of alternative labeling methods. For instance, QD photostability underlies their prime utility, but this paper like others obtains high-resolution tracks based on Atto dyes. The ~10 nm precision achieved in typical experiments here is only slightly better than for the best dyes; much of that error stems inevitably from protein motion and other aspects of the imaging strategy not from error that could be overcome with more signal photons. Thus, the impact of the paper is undermined by the fact that the authors present no particularly compelling reason to use QDs over other label types.

As previously mentioned, we reported in this paper unusually good photostability for the Atto647N and the CF633 dyes. We consider that this is confirmatory of our sQD data. We got 3D high-precision data, which is significantly better than any other data. Note that achieving 3D data requires more photons than simply 2D, so the extra photons are desired. Furthermore and most importantly, sQDs still have much better photostability as compared with any organic fluorophores, and that enabled us to perform imaging experiments where we tracked the same single particle for over 15 minutes. We now show data on this topic in Figure 7 and Figure 7—figure supplement 1,Figure 7—figure supplement 3. The data clearly show that nanodomains are (surprisingly) stable.

Fundamentally, the data in the paper are weak in that it appears that most results were obtained from single example neurons. For instance, the 2566 trajectories from the one bQD neuron in Figure 6 are by no means independent measurements.

We are sorry for the confusion and for not being clear in previous submission. The data is from independent measurements made from a minimum of 6 different areas per coverslip and 4 independent culture preparations. So the 2566 trajectories mentioned, in fact, are truly independent measurements.

Related, the idea that a change in diffusion from 1 to 3 days transfection is at best indirect evidence of cross-linking. Won't the difference in α vs β peak height difference be extremely sensitive to the proportion of the imaged area of the dendrite occupied by synapses? Isn't a difference thus expected during this developmental period of intense synaptogenesis? Because this varies widely between cells, this can only be addressed more systematically than is done here.

The data was taken on many different cells and neurons. Hence, the difference between day 1 and day 3 is robust, evidently independent of synaptogenesis. Furthermore, it is dependent on the particular label--e.g. bQD showed major differences, whereas sQD showed little difference, and organic dyes showed no difference. Even though the difference is truly independent of synaptogenesis, we did deemphasize the possibility of cross-link and suggest it as one of the possibilities.

If the major point is to compare large and small QDs, can't the two probes be applied together and imaged in the same neurons at the same time? This would give much higher confidence in any differences observed.

To use two qdots, they should be spectrally separated, but it is not easy to completely separate them because only a red color is free to use qdots due to synapse imaging with mGEOS. Furthermore, there is always the question if a bQD and a sQD enter in the same cell, what effect does two of them have (particularly if they both enter the same synapse).

Reliance on overexpressed AMPARs undermines the potential relevance of the conclusion. We do not know how the expressed receptors recapitulate, avoid, or interfere with the expression pattern of endogenous receptors.Problems associated with the size of the quantum dot and the ability of a multi-liganded label to oligomerize should not be conflated. Small, polyvalent labels can cross-link receptors yet presumably can enter the synapse well.The paper is riddled with grammatical and language errors and is far from suitable for publication. It will require extensive editing before any new submission.

We modified and answered all the questions. We added the data with native AMPARs which show very similar to the result with overexpression. We explained why we select to transfect GluA2 and Homer1c which are minimized the affect of overexpression. In terms of cross-linking, as answered above, we deemphasized the possibility of cross-linking. (Whatever the reason for the time-dependence with bQD, it is completely gone with the sQDs and the organic fluorophores.) In addition, in order to show the advantage to use or develop small qdots, we added the new data which is the long-term measurement of AMPA receptors as in Figure 7 and Figure 7—figure supplement 1,Figure 7—figure supplement 3. This shows the dynamics of the nano-domain of AMPA receptors at a single receptors level.

Reviewer #3:The purpose of this study is to determine whether AMPARs are delivered to the synapse by lateral diffusion or exocytosis. This study uses extracellularly tagged GluA2 to track the movement of AMPARs and determine their ability to localize at synapses. They use different sized tags to determine what effect this has on synaptic trafficking. Not surprisingly they find that large QDs impede synaptic trafficking more than small QDs. It is unclear to me how the authors use this data to argue that synaptic AMPARs are delivered by exocytosis. The logic escapes me.

We apologize for the miscommunication. The statement about AMPAR exocytosis we made previously in the Abstract has been eliminated. We had tried to raise the point that the rate of AMPAR exocytosis may contribute to the rate of AMPAR entry into PSDs. A brief discussion of what is rate limiting with respect to AMPAR entry into PSDs is in the Discussion of the new version of the manuscript.

There has been a lot of work using QDs carried out by the Choquet lab. The authors fail to discuss this work adequately. Do they agree with this work or not. The author's cite a few old papers to support their conclusion that there are few nonsynaptic AMPARs. However, there are many papers that unequvically show the existence of endogenous AMPARs on extrasynaptic mambranes. For instance, there is EMgold (e.g., Tao-Cheng et al., 2011), outside out dendritic patches (e.g., Andrasfalvy et al., J. Physiol. 552: 35, 2003) and 2P glutamate uncaging on dendritic shafts (e.g., Soares et al., J. Neurosci. 2013).

We have rewritten the Discussion to make it clearer about what aspects of the work from the Choquet lab we agree with and what we disagree with. Briefly, our main disagreement with the Choquet lab is the interpretation of the work using bQD-tagged AMPARs. We obtain basically the same results tracking bQD-tagged AMPARs as the Choquet lab. They have interpreted the results as indicating that most AMPARs are located outside of PSDs in a highly mobile pool. We disagree with this interpretation because additional experiments performed in our manuscript here, using either sQD-tagged AMPARs or AMPARs tagged with even smaller organic fluors, gave very different results. We have interpreted all of the results as indicating that most AMPARs are located within of PSDs showing constrained diffusion in AMPAR nanodomains. Furthermore, we interpret the opposite results obtained with bQD-tagged AMPARs as indicating that the bQDs act to hinder AMPAR entry into PSDs.

We are not claiming that all AMPARs are found in PSDs. There is clearly ~10% of the AMPARs that are extrasynaptic and some of these are likely highly mobile AMPARs diffusing from site of exocytosis to PSDs and from PSDs to sites of endocytosis. As we explain in the Discussion, our data in good agreement with the papers cited by the reviewer and “the existence of endogenous AMPARs on extrasynaptic membranes.”

[Editors' note: the author responses to the re-review follow.]

The manuscript has been improved but there are some remaining issues that need to be addressed before a final decision can be made, as outlined below:1) Overall, the presentation is rather impenetrable. The key comparisons are spread over the paper, presented confusingly, and are diluted by unrelated details. A tendency to state important results as true without quantitative support is still also a weakness here. A serious effort should be made to improve the flow of the paper and presentation of the data and conclusions.

We have made a number of detailed, but important changes, which we hope will make the paper easier to read. In particular, we now define the term “nanodomain” as sub-synaptic domain: they are *not* at all equivalent to “synaptic” domains, where the AMPARs tend to diffuse in. This is the same definition used by Nair et al.,2013, Figure 7, may have added to this confusion because it wrongly implied that these nanodomains were sometimes the same size of the synapses. (They are smaller.) We have also defined “slots” according to their original definition by Malinow. Figure 5 is also somewhat changed. We have hopefully clarified the conclusions of the paper, using these more-clear definitions.

2) Data was added in Figure 5, but 5D vs 5E are so obviously different that it undermines the conclusion that they are the same in the way the authors stress. If these images were analyzed in the same way, how could one distribution cut off abruptly at ~1 μm where the other extends smoothly to 2 um? Plus, there's obviously a prominent peak at ~0.4 μm in 5D that is absent in 5E. Isn't this critical to interpreting the distribution of receptors or the performance of the labeling strategies? Sure, the general point holds and supports the general conclusion, but to baldly state that they are the same when they in fact are quite different really suggests that the authors haven't evaluated this question very critically.

The data in Figure 5 show that the distances between Homer1 and the receptors when analyzed using the same software but using different fluorophores and live versus fixed neurons. As we described in subsection “Using sQDs for longer measurements of AMPARs in PSDs” of the main text, Atto647N created more background noise than did CF633, perhaps caused by the charge and hydrophobicity of the two dyes: Atto647N is positively charged and hydrophobic, whereas CF633 is negatively charged and hydrophilic (Zanetti-Domingues et al., 2013). Also, Figure 5 shows measurement taken of live neurons, while Figure 5 shows measurement from fixed neurons. We believe the broader distribution shown in Figure 5 is the result of the dyna mics inherent in live neurons, with Atto647N showing a bit broader distribution of ~ 0.3 or 0.4 due to the noisiness of the data. In order to better show the substantial similarity between Figure 5 and the results obtained from cleaner data and fixed neurons shown in Figure 5, we have changed Figure 5 to show the results using CF633 as the main figure and the results using the Atto647N dye in inset. We hope to have clarified this issue by making the change to Figure 5 and by adding explanation to the text.

3) Figure 7 was added, which is interesting and a good use of the QDs. The volume illustrations in panels C, D, E are nearly uninterpretable, though, since the overlays become almost black. Is there evidence that the receptor imaged in different epochs really is the same receptor? Surely there must be some exchange, increases, or decreases in number over 15 min. More seriously, there is little attempt to use the approach to evaluate some of the key questions that it raises, and it boils down mostly to a couple of anecdotes rather than a serious contribution.

In response to your comment, we have increased the brightness of the images shown in Figure 7 to improve the quality particularly in the overlay areas. We have changed Figure 7 substantially so it is clear what is the movement due to the synapse moving (which is small) and what is the size of the nanodomain. It is now clear that the nanodomain is just a small segment of the synapse. In order to clarify that we observed the same receptors over 15 minutes and that they were not jumping around, we have also added Figure 7—figure supplement 3. For this experiment, we used a very low concentration of qdots to label receptors, and found that we were able to observe the total distribution of labeled qdots, tracking each individual receptor throughout the entire time frame. There was almost no change in the position of receptors during each 30 second period as shown in Figure 7, and there was no change of the position of labeled receptors in any 5 minute period, as shown in Figure 7—figure supplement 3. This indicates that we observed the same receptor throughout the measurement period. We agree that there must be exchange, increase, or decrease in the number of receptors over time, but it is not very clear yet how fast and frequently the number of receptors changes. Our observation implies that in the basal condition, there is minimal change in the number of AMPARs. The ability to see a single receptor over an extended period of the time shows the superiority of small qdots to observe the dynamics of post-synaptic receptors after LTP or LTD induction.

4) Figure 7 also provides a new but confusing operational definition of a nanodomain (paragraph three in subsection “Using sQDs for longer measurements of AMPARs in PSDs”). A nanodomain in Nair and MacGillavry is a delineated subsynaptic region where receptors are denser than elsewhere in the synapse. Through most of this paper, "nanodomain" is used pretty much synonymously with "synapse" since there's no attempt to distinguish subregions of the synapse and no effort at quantifying high or low areas of receptor density. Indeed, the size of the "nanodomain" in 7f is probably close to or equal to the entire synapse. Then in 7 to suggest that where one receptor is present for long periods is the same as one of those other nanodomains is confusing and I believe premature. I think it would be of interest to determine whether further analysis of the sQD tracking can help relate these terms to one another, but that surely would be many experiments and is well beyond the scope of this paper; instead, please avoid the confusion entirely and not use "nanodomain" so liberally particularly in this section of the paper.

We apologize for this confusion. We do not intend to use nanodomain as a synonym for “synapse”. As the editor pointed out, a nanodomain means the sub-synaptic region where receptors are highly clustered, but it also means the region where receptors are constrained by scaffold (or slot) proteins. In this paper, we used “nanodomains” as the region where a receptor was constrained. We have clarified our use of the term, as discussed in the main text of the letter above. In this study, we focused on observing the dynamics of a single receptor. Although it is possible to analyze the density of fixed neurons as added in Figure 5—figure supplement 4 (the histogram of the AMPAR density), we were not concerned with the density of receptors in this study. Even with fixed neurons, it is not easy to measure the absolute total number of labeled receptors. Fundamentally, when getting super-resolution images, the blinking of dyes is not a single event for each fluorophore and it is not possible to label 100% of the receptors, and it is not relevant to the aim of this research. Hence, we avoid discussing the density of receptors, and instead clarify our definition of nanodomains.

5) There is no definition of "constrained" diffusion in the paper, either mathematically or otherwise. Surely slow is not the same thing as constrained, as implied in subsection “AMPAR-labeling: Diffusion constants and trajectory range”? Don't the authors want to distinguish between "confined" and slow, and to measure the confinement radius? Or is "constrained" just defined as being "pretty slow and pretty short range"?

Our definition of "constrained" diffusion is that diffusing AMPARs are restricted to a subdomain by interactions with scaffold or slot proteins. Hence, we added this to subsection “AMPAR-labeling: Diffusion constants and trajectory range “, with references (Shi et al., 2001; Opazo, Sainlos and Choquet, 2012). We certainly want to distinguish between confined and slow. This is the reason why we plotted both the diffusion coefficient and the trajectory range of AMPARs in the two-dimensional heat map. In the two-dimensional heat maps, the lower left area shows confined receptors.

6) The first conclusion in the Discussion (receptor distribution around spines) is not even dealt with directly in the paper; no effort is made to measure density on spines, and no number is even given for the total proportion of receptors outside synapses-only the proportion of those that are outside the synapse but within 2 um.

As we mentioned in response to comment #4 above, we measured the dynamics of receptors at the single molecule level, so that this measurement is not a way of analyzing the density of receptors on spines. As we stated, it was previously thought that the extra-synaptic AMPA receptors are the main pool for providing receptors to the synapses, but our data suggests that the previous finding of large pools of extra-synaptic receptors was an artifact due to big size qdots. Indeed, we reproduced those results with large qdots, which we clarified this in subsection “AMPARs labeled with bQD coupled with SA are largely highly mobile and extrasynaptic”. Then in subsection “AMPARs labeled with sQD by different linkers (SA, GBP, or mSA)”, we clarified that we did not find large pools extra-synaptically when we used small qdots. In each case, our summary of results with bQDs and sQDs is followed by explanation of our experimental methodology and results.

7) There is an extensive literature using FRAP and optical antagonists to deduce receptor mobility, and this should be evaluated in light of the current results as well. FRAP results have shown mobile fractions ranging from 10 to 90% and have been key support for the idea that receptors continually exchange and diffuse into the synapse during LTP.

Yes, we are aware of this previous literature. Almost all studies using FRAP used expressed proteins such as AMPAR subunits with fluorescent proteins tags. A concern that we have with these studies is the difficulty in distinguishing whether the proteins are located in an intracellular organelle or on the cell surface. Thus, we think it is not a reliable method to measure the dynamics of surface AMPARs. For readers, we added more references in the third paragraph of the Introduction section.

8) The final paragraph of the Discussion contains many premature statements.

We apologize for a lack of clarity about the definition of AMPAR PSD slots and some of our statements about these slots. We have rephrased the relevant parts of the Abstract, Introduction and the last paragraphs of the Discussion, to make this issue clearer. Of most relevance is the hypothesis of other workers that the accumulation of mobile, extrasynaptic AMPARs is caused by the occupancy of AMPAR slots in PSDs. Our findings that this pool of extrasynaptic AMPARs is absent indicates that previous observations of a large mobile extrasynaptic AMPAR pool is an artifact of using big quantum dots.

– The PSD slot hypothesis long predates Opazo, Sainlos and Choquet, 2012. It stemmed from electrophysiological observations following molecular manipulations, not from imaging results.

We agree that the PSD slot hypothesis predates Opazo, Sainlos and Choquet, 2012 and have added additional references to reflect this fact.

– The slot hypothesis even as stated in Opazo does not predict that AMPARs cannot enter the PSD-it merely states that there is no way for synapses to anchor and retain them if they do enter.

We completely agree with the reviewer and have changed the writing so this no longer is an issue.

– It is unclear why a small pool of extrasynaptic receptors suggests that there are or are not slots.

We apologize for the confusion and have rephrased the Abstract so that there is no longer reference to the existence of slots.

– The presence of clustered receptors in the synapse also does not seem to bear on whether a saturable molecular complex is involved in retaining them.

Again, we agree with the reviewer and hope that the rewriting has addressed this concern.

– If the authors wish to argue that mobility itself is a sign that slots do not exist, they need to make that argument more carefully and thoroughly. Isn't a receptor bound to almost any component of the PSD likely to be nearly completely immobilized?

As stated immediately above, we do not want to argue whether slots do exist or not

– What's the difference between a slot and a place within the synapse where a receptor is constrained from leaving?

We simply suggest that sQDs show that receptors are, contrary to previous research with bQDs, in the synaptic regions. See subsection “AMPARs labeled with sQD by different linkers (SA, GBP, or mSA)”. We do not know nor claim that the reason is that there are slots or other places within the synapse that constrains the receptors

9) A comment from the original second reviewer seems to have been somewhat poorly answered: "Fundamentally, the data in the paper are weak in that it appears that most results were obtained from single example neurons. For instance, the 2566 trajectories from the one bQD neuron in Figure 6 are by no means independent measurements.". The authors reply that "The data is from independent measurements made from a minimum of 6 different areas per coverslip and 4 independent culture preparations. So the 2566 trajectories mentioned, in fact, are truly independent measurements." This reply is not fully correct. The 2566 trajectories are not truly independent from each other – surely at least some trajectories come from the same series of images (movies). The authors should make more efforts to point out how many trajectories were analyzed per movie, and how many movies they analyzed, from how many coverslips, and from how many independent cultures.

We added this information in figure legends of Figure 3–Figure 6, e. g., “6 cells, 4 independent dissociation, 1 dissociation per week”